# S$^2$Q-VDiT: Accurate Quantized Video Diffusion Transformer with Salient Data and Sparse Token Distillation

**Weilun Feng**[1,2]*, **Haotong Qin**[3]*, **Chuanguang Yang**[1]†, **Xiangqi Li**[1,2], **Han Yang**[1], **Yuqi Li**[1],
**Zhulin An**[1]†, **Libo Huang**[1], **Michele Magno**[3], **Yongjun Xu**[1]

[1]State Key Laboratory of AI Safety, Institute of Computing Technology, Chinese Academy of Sciences
[2]University of Chinese Academy of Sciences    [3]ETH Zürich

{fengweilun24s,yangchuanguang,lixiangqi24s,anzhulin,xyj}@ict.ac.cn
{haotong.qin,michele.magno}@pbl.ee.ethz.ch,    {yuqili010602,www.huanglibo}@gmail.com

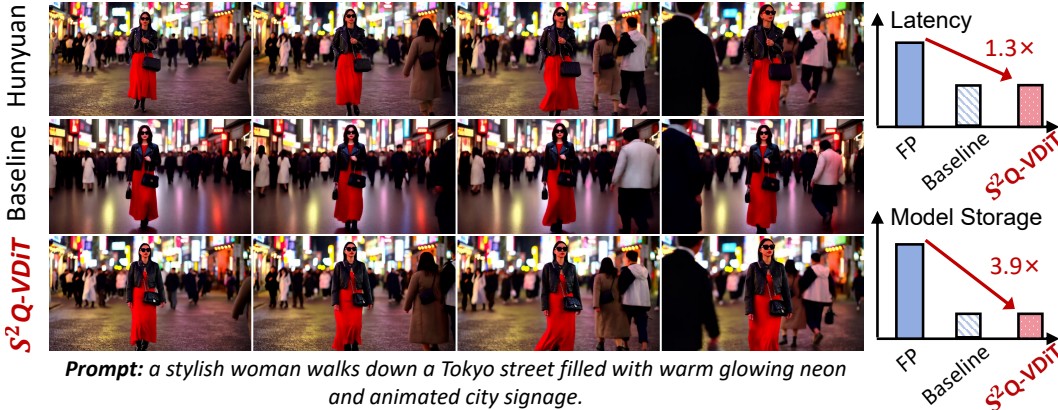

**Prompt:** *a stylish woman walks down a Tokyo street filled with warm glowing neon and animated city signage.*

Figure 1: We present S$^2$Q-VDiT, a post-training quantization method for video diffusion transformers. We quantize HunyuanVideo [24] to 4-bit weights and 6-bit activations without compromising visual quality. S$^2$Q-VDiT can further achieve 3.9× model compression and 1.3× inference acceleration.

## Abstract

Diffusion transformers have emerged as the mainstream paradigm for video generation models. However, the use of up to billions of parameters incurs significant computational costs. Quantization offers a promising solution by reducing memory usage and accelerating inference. Nonetheless, we observe that the joint modeling of spatial and temporal information in video diffusion models (V-DMs) leads to extremely long token sequences, which introduces high calibration variance and learning challenges. To address these issues, we propose **S$^2$Q-VDiT**, a post-training quantization framework for V-DMs that leverages **S**alient data and **S**parse token distillation. During the calibration phase, we identify that quantization performance is highly sensitive to the choice of calibration data. To mitigate this, we introduce *Hessian-aware Salient Data Selection*, which constructs high-quality calibration datasets by considering both diffusion and quantization characteristics unique to V-DMs. To tackle the learning challenges, we further analyze the sparse attention patterns inherent in V-DMs. Based on this observation, we propose

---

*Equal contribution.
†Corresponding authors: Zhulin An, anzhulin@ict.ac.cn; Chuanguang Yang, yangchuanguang@ict.ac.cn

39th Conference on Neural Information Processing Systems (NeurIPS 2025).

*Attention-guided Sparse Token Distillation*, which exploits token-wise attention distributions to emphasize tokens that are more influential to the model's output. Under W4A6 quantization, S$^2$Q-VDiT achieves lossless performance while delivering $3.9\times$ model compression and $1.3\times$ inference acceleration. Code will be available at `https://github.com/wlfeng0509/s2q-vdit`.

# 1   Introduction

In recent years, diffusion transformer [39] has emerged as a powerful generative paradigm, demonstrating remarkable performance across diverse domains such as image synthesis [6, 26, 9, 57], audio generation [15], and increasingly, video generation [37, 35]. Among these, video diffusion models (V-DMs) [58, 24] represent a new frontier by extending the spatial generative capabilities of image diffusion models (I-DMs) into the spatial-temporal domain, enabling high-quality video synthesis from textual prompts.

However, the transition from image to video generation introduces substantial computational challenges, primarily due to the exponential growth in token count introduced by the temporal dimension [35, 58, 24]. These memory and compute demands become particularly severe in large-scale video generation models [35, 58, 24], which contain up to billions of parameters, where each input consists of thousands or even tens of thousands of tokens. To enable efficient deployment of such models in resource-constrained environments, post-training quantization (PTQ) [32, 52, 20, 5] has become a widely adopted approach. PTQ compresses the pre-trained models into low-bit representations without modifying the model weights, relying only on a small dataset to calibrate quantization parameters with only hours on a single GPU [51, 28].

While PTQ has proven effective for I-DMs [30, 45, 54], directly applying it to V-DMs leads to substantial performance degradation [2, 62]. Prior works [2, 54, 62] have sought to improve V-DMs' quantization performance primarily from the perspective of quantizer design. In this paper, we delve deeper into the PTQ challenges specific to V-DMs, focusing on calibration data and optimization methods.

We identify that the long token sequences characteristic of V-DMs significantly constrain the number of calibration samples (e.g., thousands for I-DMs vs. only dozens for V-DMs under equal computational budgets). Under such limited budgets, quantization performance becomes highly sensitive to the selection of calibration samples. Existing methods [54, 2, 62] typically employ random or uniform sampling strategies, which work reasonably well for I-DMs but fail to generalize well to only dozens of data for V-DMs. Moreover, we observe that V-DMs exhibit sparse attention patterns across all tokens. Current PTQ optimization frameworks [54, 30] treat all tokens equally during loss alignment between full-precision and quantized models. However, this uniform treatment is suboptimal for long token sequences, where only a small subset of tokens significantly impacts the final output. These observations highlight two fundamental challenges in PTQ for V-DMs: (1) the absence of a principled method for selecting calibration samples, and (2) the inefficiency of uniform token treatment during optimization, despite the varying importance of tokens.

To address these limitations, we propose **S$^2$Q-VDiT**, a post-training quantization framework tailored for V-DMs, built upon **S**alient data selection and **S**parse token distillation. An overview of the proposed framework is illustrated in Fig. 2. First, we introduce *Hessian-aware Salient Data Selection*, which constructs calibration datasets by jointly assessing diffusion informativeness and quantization sensitivity. We define a unified metric to quantify sample's saliency to the denoising process and its sensitivity to quantization perturbations. Second, we present *Attention-guided Sparse Token Distillation*, a technique that leverages the inherent sparsity of spatial-temporal attention in V-DMs. Rather than treating all tokens equally during optimization, we reweight quantization losses based on token-wise attention distribution, allowing the model to focus more on the impactful representations.

Our main contribution can be summarized as follows:

- We empirically identify that V-DMs suffer from high calibration data variance in quantization performance. We propose *Hessian-aware Salient Data Selection*, which jointly considers diffusion informativeness and quantization sensitivity to construct effective calibration datasets.

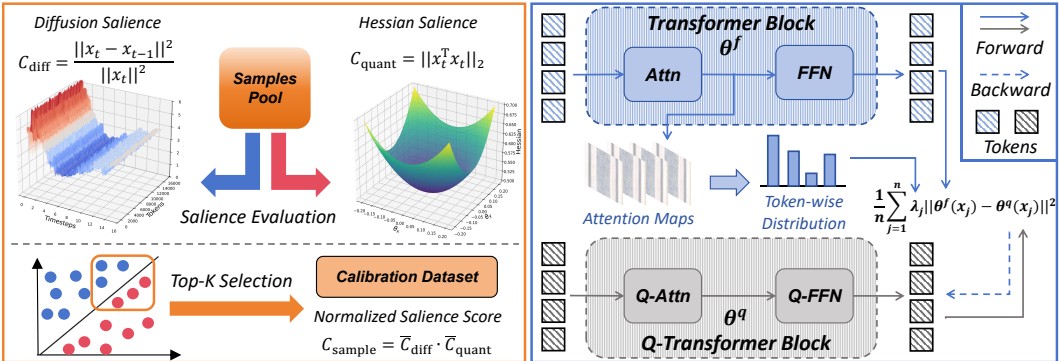

Figure 2: Overview of S$^2$Q-VDiT. The framework includes Hessian-aware Salient Data Selection (SDS) for constructing calibration dataset and Attention-guided Sparse Token Distillation (STD) for block-wise optimization.

- We introduce *Attention-guided Sparse Token Distillation*, a method that leverages the inherent sparsity in spatial-temporal attention of V-DMs. We reweight the quantization loss of different tokens by measuring token-wise attention distribution. This enables the model to focus more on the impactful representations during optimization.

- Extensive experiments on large-scale video diffusion transformers with 2B to 13B parameters demonstrate that our **S$^2$Q-VDiT** consistently outperforms existing PTQ baselines, achieving state-of-the-art performance under all quantization settings.

## 2   Related Works

Diffusion models[46, 17] have demonstrated strong generative capabilities in video generation tasks. However, up to billions of parameters [35, 49, 58, 24] pose major challenges for deployment in resource-constrained environments. Quantization has emerged as a widely adopted solution for model compression and acceleration [40, 14, 3, 21, 25]. A growing body of work has explored post-training quantization (PTQ) for diffusion models, particularly focusing on U-Net-based architectures [30, 45, 16, 18, 29, 63]. For the Diffusion Transformer architecture specifically, recent methods [54, 2, 10] have made further explorations from the perspective of data distribution and architecture characteristics on quantization behavior. To address performance degradation at ultra-low bits, several quantization-aware training approaches have been proposed [64, 31, 36, 65, 8, 11]. While effective, these methods typically require extensive training time and large-scale datasets, making them less practical in many scenarios.

Despite these advances, most existing quantization research remains focused on image diffusion models (I-DMs), with limited exploration of video diffusion models (V-DMs). ViDiT-Q [62] and Q-DiT [2] have made the first explorations on the quantization of V-DMs. Q-DiT [2] introduces automatic quantization granularity allocation for fine-grained quantizer selection. ViDiT-Q [62] proposes static-dynamic quantization strategy to enhance quantization accuracy. While these approaches improve performance from different perspectives, they primarily focus on quantization granularity and quantizer design. In this paper, we tackle V-DM quantization from a new angle—calibration data quality and optimization strategy. Our method achieves lossless performance on various large-scale video diffusion transformers from 2B to 13B.

## 3   Methods

### 3.1   Preliminary

**Video Diffusion Transformer.**   Diffusion transformers [39] predict the target using the representation of multiple tokens $X \in \mathbb{R}^{n \times d}$ where $n$ and $d$ represent the number of tokens and feature dimension, respectively. For image diffusion models (I-DMs) [42, 26], $n = s$ accounts for spatial

tokens. But for video diffusion models (V-DMs) [58, 24, 37], $n = s \times t$ incorporates the temporal dimension $t$. This significantly increases the token count per sample (e.g., $t = 49$ for a 6-second video at 8 FPS), resulting in heightened memory consumption and greater optimization complexity.

**Post-training Quantization.** Quantization maps the model weights and activation to low-bit integers for acceleration and memory saving. For a float vector $x$, the symmetry quantization process can be formulated as:

$$x_{\text{int}} = \text{clamp}(\text{round}[\frac{x}{\Delta}], -2^{N-1}, 2^{N-1} - 1), \ \Delta = \frac{\max(abs(x))}{2^{N-1} - 1} \tag{1}$$

where $N$ is the quantized bit, $round(\cdot)$ is the round operation, and $clamp(\cdot)$ constrains the value within integer range $[-2^{N-1}, 2^{N-1} - 1]$. Among quantization methods, post-training quantization (PTQ) is a more efficient method that only calibrates quantization parameters using a small calibration dataset $D_{\text{calib}}$ without altering model weights. According to common practice [1, 55, 62], the quantization loss is expressed as:

$$\mathcal{L}_{\text{quant}} = \mathbb{E}_{X \sim D_{\text{calib}}}[||\theta^f(x) - \theta^q(x)||^2], \tag{2}$$

where $\theta^f$ and $\theta^q$ denote the full-precision and quantized model parameters, respectively. $D_{\text{calib}} \in \mathbb{R}^{N \times n \times d}$ where $N$ denotes the sample number in $D_{\text{calib}}$. Due to the limitations in computing resources and long token sequences in V-DMs, the calibration sample size $N$ is smaller than that in I-DMs, leading to higher variance in data representation. This variance is further exacerbated by the diverse text prompts and different denoising timesteps present in the diffusion models.

## 3.2 Hessian-aware Salient Data Selection

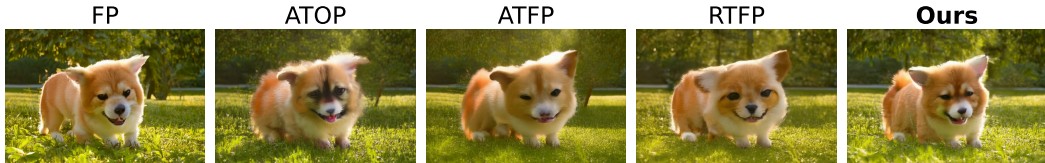

| FP | ATOP | ATFP | RTFP | **Ours** |

Figure 3: Visualization of different calibration data on CogVideoX-2B. We compare our proposed method with All Timesteps from One Prompt (ATOP), All Timesteps from Five Prompts (ATFP), and Random Timesteps from Five Prompts (RTFP). Our method has better generation quality.

**Observation 1.** *Calibration sample selection methods result in high variance of the quantized model performance.*

In line with we discussed in Sec. 3.1, we observed that under constrained calibrated data size, different samples have significant differences in the final model performance as shown in Fig. 3 and Fig. 6a. However, the sample selection method for V-DMs post-training quantization has not been thoroughly explored. Therefore, we hope to evaluate the importance of different data for V-DMs. To address this issue, we propose evaluating sample utility along two dimensions that naturally exist in the quantization of diffusion models: contribution to the diffusion process and sensitivity to quantization.

Prior work on timestep distillation [43, 44] and caching [33, 22] indicates that skipping certain consecutive timesteps has limited impact on output quality, suggesting varying information content across different timesteps. Based on this insight, we measure the salient information of timestep $t$ for the whole denoising diffusion process by calculating the contribution of two consecutive timesteps latent representation. Given all candidate data among all the diffusion timesteps $[x_1, x_2, \cdots, x_T]$ where $T$ is the total denoising timesteps defined in the pretrained models. We define the diffusion salience as:

$$C_{\text{diff}} = \frac{||x_t - x_{t-1}||^2}{||x_t||^2}, \tag{3}$$

where $x_t$ stands for the denoised feature of timestep $t$. A higher $C_{\text{diff}}$ value denotes more informative denoising steps, while a lower $C_{\text{diff}}$ value indicates that the contained information largely overlaps with the previous timestep. $C_{\text{diff}}$ naturally measures the saliency of different timesteps during the diffusion denoising process. By focusing on the salient data, we can better approximate the distribution of the entire diffusion process and achieve better performance.

We then consider the quantization of weight $W$ and its quantized version $\hat{W} = W + \Delta$, the quantization loss that jointly considers the input $X$ can be be approximated using a Taylor expansion:

$$\mathbb{E}[||XW^\top - X\hat{W}^\top||^2] = \mathbb{E}[||XW^\top - X(W + \Delta)^\top||^2]$$
$$\approx \Delta g^X + \frac{1}{2}\Delta H^X \Delta^\top, \quad (4)$$

where $g^X$ is the gradient and $H^X$ is the Hessian matrix. Using $g^X = 0$ for a well-trained model provided in [32, 59] and $H^X = \mathbb{E}[2X^\top X]$ provided in [13], Eq. (4) can be further simplified to:

$$\mathbb{E}[||XW^\top - X\hat{W}^\top||^2] \approx \mathbb{E}[\Delta(X^\top X)\Delta^\top], \quad (5)$$

where Hessian matrix $X^\top X$ is given by Levenberg-Marquardt approximation [12, 38]. The Hessian matrix represents the inherent perturbation ability of sample $X$ to the quantization process, which leads us to define quantization salience as:

$$C_{\text{quant}} = ||x_t^\top x_t||_2, \quad (6)$$

where a larger $C_{\text{quant}}$ denotes that $x_t$ is more sensitive to the quantization process due to the property of the Hessian matrix [13, 12, 59]. By focusing on the quantization-sensitive samples, we can further relieve the bridge between the original data distribution and quantization operations, making the quantized model more robust and perform better.

To jointly emphasize diffusion informativeness and quantization sensitivity, we apply min–max normalization over the candidate calibration pool $\mathcal{D}_{\text{calib}}$:

$$\overline{C}_{\text{diff}}(x_t) = \frac{C_{\text{diff}}(x_t) - C_{\text{diff}}^{\min}}{C_{\text{diff}}^{\max} - C_{\text{diff}}^{\min}}, \quad \overline{C}_{\text{quant}}(x_t) = \frac{C_{\text{quant}}(x_t) - C_{\text{quant}}^{\min}}{C_{\text{quant}}^{\max} - C_{\text{quant}}^{\min}}, \quad (7)$$

where $C_{\text{diff}}^{\min}$, $C_{\text{diff}}^{\max}$, $C_{\text{quant}}^{\min}$, and $C_{\text{quant}}^{\max}$ denote the mininum value and maxminum value of all $C_{\text{diff}}(\cdot)$ and $C_{\text{quant}}(\cdot)$ respectively. The unified salience score is then defined as the product:

$$C_{\text{sample}}(x_t) = \overline{C}_{\text{diff}}(x_t) \cdot \overline{C}_{\text{quant}}(x_t) \leq \left(\frac{\overline{C}_{\text{diff}}(x_t) + \overline{C}_{\text{quant}}(x_t)}{2}\right)^2, \quad (8)$$

by the Arithmetic–Geometric Mean inequality [67] which ensuring the score is maximized only when both normalized metrics are high. This mutual-salience product metric inherently penalizes samples that are only strong on one dimension, aligns with mutual-information objectives, and yields a more strong, robust calibration set.

### 3.3 Attention-guided Sparse Token Distillation

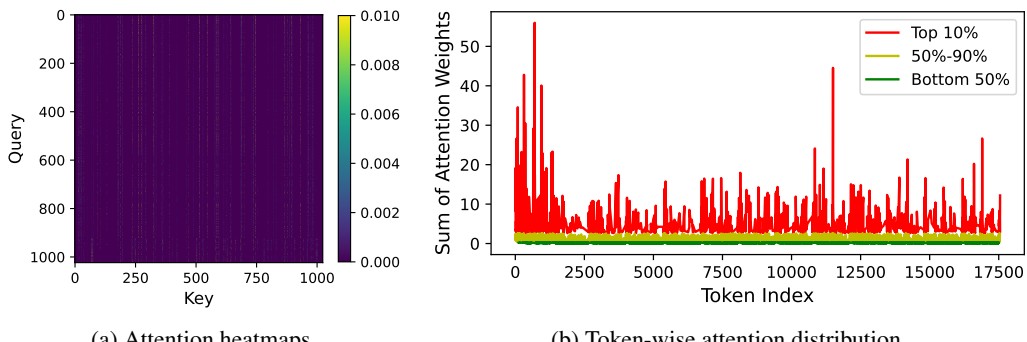

| (a) Attention heatmaps. | (b) Token-wise attention distribution. |

Figure 4: Visualization of sparse attention patterns in CogVideoX-2B block-10. In (4a), fewer columns have significantly higher weights. In (4b), only 10% of tokens have larger attention weights.

**Observation 2.** *The fully spatial-temporal attention in V-DMs exhibits certain sparse patterns, suggesting that only subsets of tokens notably impact the model output.*

Table 1: Performance of 4-bit weight and 6-bit activation quantization on text-to-video generation under VBench evaluation benchmark suite. We evaluate on Imaging Quality (IQ), Aesthetic Quality (AQ), Motion Smoothness (MS), Dynamic Degree (DD), Background Consistency (BC), Subject Consistency (SuC), Scene Consistency (ScC), and Overall Consistency (OC). Higher (↑) metrics represent better performance. **Bold**: the best result.

| Model | Method | IQ | AQ | MS | DD | BC | SuC | ScC | OC |
|---|---|---|---|---|---|---|---|---|---|
| CogVideoX-2B | FP | 58.69 | 55.25 | 97.95 | 50.00 | 96.40 | 94.30 | 33.79 | 25.91 |
| | Q-DiT | 48.63 | 47.63 | 98.08 | 19.44 | 95.30 | 92.15 | 23.84 | 24.00 |
| | PTQ4DiT | 42.91 | 45.49 | **98.48** | 5.56 | 95.65 | 92.85 | 17.88 | 21.15 |
| | SmoothQuant | 44.60 | 44.33 | 98.22 | 9.72 | 95.62 | 92.04 | 18.60 | 21.20 |
| | Quarot | 51.89 | 48.48 | 97.49 | 31.94 | 95.61 | 93.01 | 22.97 | 23.57 |
| | ViDiT-Q | 51.94 | 48.06 | 97.47 | 33.33 | 95.54 | 92.87 | 22.17 | 23.69 |
| | **S²Q-VDiT** | **55.49** | **53.74** | 98.10 | **40.28** | **96.05** | **94.16** | **32.70** | **25.19** |
| CogVideoX-5B | FP | 61.80 | 58.88 | 97.61 | 72.22 | 95.56 | 94.63 | 45.28 | 26.46 |
| | Q-DiT | 49.94 | 50.18 | 97.03 | 43.06 | 95.52 | 91.58 | 29.65 | 24.49 |
| | PTQ4DiT | 43.54 | 42.70 | 97.77 | 4.17 | 96.70 | 93.32 | 10.93 | 21.75 |
| | SmoothQuant | 39.50 | 36.92 | **97.88** | 6.94 | 96.39 | 92.28 | 23.11 | 18.19 |
| | Quarot | 43.95 | 44.81 | 97.33 | 31.94 | 96.58 | 92.27 | 20.93 | 22.34 |
| | ViDiT-Q | 48.87 | 50.51 | 97.66 | 37.50 | 96.25 | 93.60 | 27.76 | 23.57 |
| | **S²Q-VDiT** | **60.75** | **56.90** | 97.46 | **58.33** | **96.76** | **94.24** | **46.66** | **26.30** |
| HunyuanVideo | FP | 62.30 | 62.49 | 99.00 | 56.94 | 98.08 | 95.30 | 33.36 | 26.85 |
| | Q-DiT | 50.23 | 48.40 | 98.95 | 40.28 | 97.14 | 94.03 | 18.46 | 14.41 |
| | PTQ4DiT | 48.31 | 50.13 | 98.26 | 19.44 | 97.95 | 94.37 | 20.19 | 19.85 |
| | SmoothQuant | 47.55 | 56.03 | 98.77 | 27.78 | 97.33 | 94.57 | 23.69 | 25.47 |
| | Quarot | 52.31 | 58.50 | 99.13 | 37.50 | 97.98 | 95.31 | 25.51 | 26.01 |
| | ViDiT-Q | 52.21 | 58.38 | 99.12 | 41.67 | 98.02 | 95.20 | 23.69 | 26.15 |
| | **S²Q-VDiT** | **58.83** | **59.62** | **99.20** | **48.61** | **98.15** | **95.57** | **33.65** | **26.91** |

Let $x \in \mathbb{R}^{n \times d}$ be the token embeddings, we can express Eq. (2) in the summation form as follows:

$$\mathcal{L}_{\text{quant}} = \frac{1}{n} \sum_{j=1}^{n} ||\theta^f(x_{j,:}) - \theta^q(x_{j,:})||^2, \tag{9}$$

where $x_{j,:}$ refers to the $j_{th}$ token in the video diffusion transformer. This loss function assumes that each token contributes equally to the overall error between the quantized and full-precision models. However, due to the spatial-temporal modeling objectives, V-DMs typically require large-scale pretraining to achieve full convergence [37, 58, 24, 56].

In the post-training quantization (PTQ) stage, only a small dataset is used to calibrate the quantization parameters, which naturally limits the model's ability to optimize from all tokens. Nevertheless, attention maps derived from V-DMs reveal that only subsets of tokens significantly influence the final output (see Fig. 4 and Appendix Sec. H). This observation aligns with prior studies on attention in V-DMs [60, 4, 61, 66], which have shown that pruning irrelevant tokens has a negligible impact on generation quality. These findings motivate a strategy that focuses learning more intensely on salient tokens while relaxing constraints on less impactful ones. Thereby enabling better convergence and improved performance even with limited calibration data.

To improve alignment between quantized and full-precision outputs, we reweight each token's contribution in the loss function based on its influence on the block output. Formally, we modify Eq. (9) to:

$$\mathcal{L}_{\text{quant}} = \frac{1}{n} \sum_{j=1}^{n} \lambda_j ||\theta^f(x_{j,:}) - \theta^q(x_{j,:})||^2, \tag{10}$$

where $\lambda_j$ denotes the weighting factor corresponding to token $x_{j,:}$. Leveraging the attention mechanism within each transformer block of V-DMs, we can obtain the complete multi-head attention

Table 2: Performance of both 4-bit weight and activation quantization on text-to-video generation under VBench evaluation benchmark suite.

| Model | Method | IQ | AQ | MS | DD | BC | SuC | ScC | OC |
|-------|--------|-----|-----|-----|-----|-----|-----|-----|-----|
| CogVideoX-2B | FP | 58.69 | 55.25 | 97.95 | 50.00 | 96.40 | 94.30 | 33.79 | 25.91 |
| | Q-DiT | 26.26 | 27.66 | 99.14 | 0 | **98.09** | **96.52** | 1.16 | 8.43 |
| | PTQ4DiT | 20.66 | 28.50 | **99.30** | 0 | 97.61 | 95.33 | 2.11 | 11.11 |
| | SmoothQuant | 29.76 | 28.31 | 98.95 | 0 | 97.62 | 94.65 | 5.31 | 9.74 |
| | QuaRot | 43.22 | 39.59 | 97.54 | 13.89 | 96.18 | 92.35 | 12.21 | 19.57 |
| | ViDiT-Q | 45.56 | 42.03 | 97.57 | 12.5 | 96.08 | 92.43 | 11.91 | 19.61 |
| | **$S^2$Q-VDiT** | **53.71** | **52.31** | 98.09 | **36.11** | 96.15 | 93.99 | **34.23** | **24.90** |
| CogVideoX-5B | FP | 61.80 | 58.88 | 97.61 | 72.22 | 95.56 | 94.63 | 45.28 | 26.46 |
| | Q-DiT | 40.80 | 33.00 | 95.71 | 36.11 | **98.26** | **96.99** | 0.22 | 1.91 |
| | PTQ4DiT | 41.48 | 28.63 | 96.38 | 0 | 97.29 | 95.09 | 0 | 7.37 |
| | SmoothQuant | 40.30 | 29.99 | 95.76 | 1.39 | 96.54 | 96.02 | 0.44 | 6.51 |
| | QuaRot | 29.41 | 35.36 | **97.77** | 15.28 | 97.23 | 92.71 | 8.36 | 15.31 |
| | ViDiT-Q | 31.95 | 36.71 | 97.09 | 15.28 | 96.37 | 93.01 | 10.85 | 16.91 |
| | **$S^2$Q-VDiT** | **58.76** | **55.35** | 97.18 | **47.22** | 96.25 | 93.69 | **36.56** | **26.02** |

map $A \in \mathbb{R}^{H \times n \times n}$ where $H$ is the number of attention heads. $A$ naturally represents the importance matrix of different tokens within each block, and $A_{h,i,j}$ denotes the attention weight $j_{th}$ token receives from the $i_{th}$ token in $h_{th}$ attention head. We use the attention map $A$ to compute $\lambda_j$ using:

$$S_j = \sum_{h,i} A_{h,i,j}, \ \lambda_j = \frac{S_j - \min(S)}{\max(S) - \min(S)}(\lambda_{\max} - \lambda_{\min}) + \lambda_{\min}, \tag{11}$$

where $\min(S)$ and $\max(S)$ denote the minimum and maximum values in all $S$ respectively. The hyperparameters $\lambda_{\min}$ and $\lambda_{\max}$ define the normalization range for token importance. Ultimately, $\lambda_j$ quantifies each token's salience and helps guide the optimization process to prioritize alignment for tokens that exert greater influence.

## 4 Experiments

### 4.1 Experimental and Evaluation Settings

**Quantization Scheme.** We employ uniform per-channel weight quantization and dynamic per-token activation quantization with channel-wise scale and rotation matrix same as prior works [2, 1, 62]. We use symmetry quantization for both weight and activation for better hardware acceleration and memory saving. We follow the block-wise post-training strategy used in prior works [30, 54, 2]. More implementation details and model settings can be seen in Appendix Sec. A.

**Evaluation Settings.** We conduct text-to-video experiment on different scale SOTA models CogVideoX-2B, CogVideoX-5B [58] and HunyuanVideo-13B [24] for better evaluation. We evaluate the performance of the quantized model using the VBench benchmark [19], which provides a comprehensive evaluation on video generation performance. Same as the prior works [2, 62], we select 8 major evaluation dimensions from VBench to ensure a thorough assessment. **We also present more experiments on EvalCrafter [34] benchmark in Appendix Sec. D**. As current works [2, 62] have achieved almost lossless performance at high bits (e.g., 6-8 bits), we evaluated the performance at more challenging and unexplored low-bit W4A6 and W4A4 settings.

**Compared Methods.** Consist with prior works [2, 62], we compare $S^2$Q-VDiT with current PTQ baseline methods. For diffusion baseline, we compare with Q-DiT [2], PTQ4DiT [54], and ViDiT-Q [62]. We further compare with strong LLM baseline, SmoothQuant [55] and QuaRot [1].

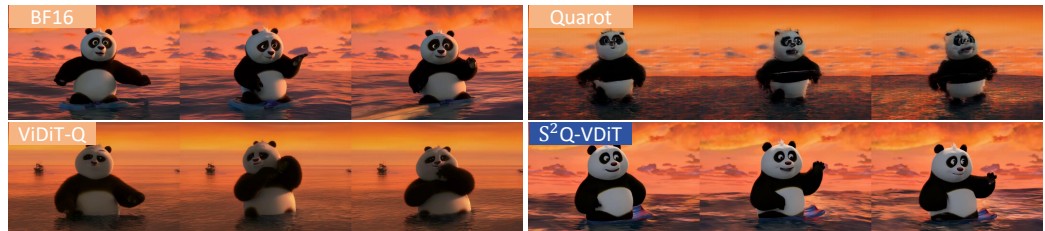

*Prompt: A panda standing on a surfboard in the ocean in sunset.*

(a) CogVideoX-5B.

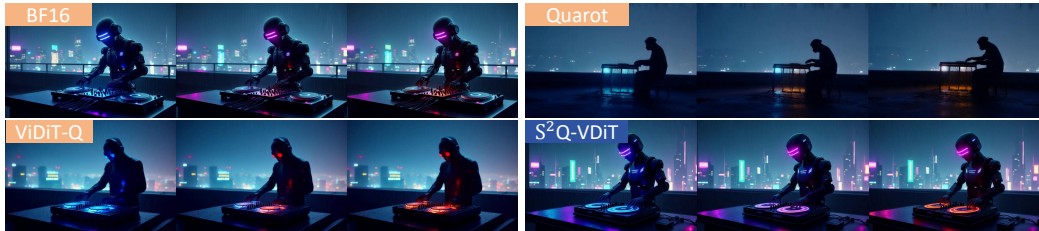

*Prompt: A robot DJ is playing the turntable, in heavy raining futuristic tokyo rooftop cyberpunk night, sci-fi, fantasy.*

(b) HunyuanVideo-13B.

Figure 5: Visual comparison on different models under W4A6 quantization setting.

## 4.2 Quantitative Comparison

We present text-to-video experiment under VBench evaluation benchmark suite in Tab. 1 and Tab. 2. **W4A6 Quantization.** In Tab. 1, we focus on relatively higher bit quantization setting of W4A6 (4-bit weight and 6-bit activation). In three different scale current V-DMs CogVideoX-2B, CogVideoX-5B, and HunyuanVideo-13B, our method outperforms all current quantization methods by a notable margin. Our $S^2$Q-VDiT achieves almost lossless performance across all eight selected dimensions. For CogVideoX-5B, $S^2$Q-VDiT even outperforms FP model with 46.66 scene consistency while other methods achieved the highest score of 29.65.
**W4A4 Quantization.** In Tab. 2, we further explored the quantization performance of V-DMs under extremely low bit W4A4 settings. It is worth noting that this is currently the first exploration under 4-bit activation quantization. In this extremely low bit setting, $S^2$Q-VDiT can still maintain 95% of the model's performance while other methods showed significant performance degradation. **Although some methods are particularly high in metrics such as SuC and BC, this is due to their almost collapsed generation quality. ViDiT-Q [62] pointed out that these metrics are particularly high on extremely collapsed methods, and maintaining performance closer to FP is better.** For CogVideoX-2B, our method achieves even lossless scene consistency of 34.23 while other methods achieved the highest score of 12.21 with almost a three times improvement.

## 4.3 Visual comparison

We present visual comparisons on different models under W4A6 in Fig. 5. Compared with the current SOTA methods QuaRot [1] and ViDiT-Q [62], $S^2$Q-VDiT has significant improvements in image quality and dynamic degree, and is lossless compared to FP models. For CogVideoX-5B, QuaRot can hardly generate clear images; ViDiT-Q lacks the ability in color richness and image details; $S^2$Q-VDiT is significantly better in color, detail, and video dynamics. For HunyuanVideo, although all methods have not significantly reduced image clarity, the semantic information of QuaRot has severely declined; the generated characters and background details of ViDiT-Q are also insufficient. $S^2$Q-VDiT maintains high quality in the details and colors of both the background and characters, and ensures the dynamic level of the video at different frame. The consistent and significant improvement on three different scales V-DMs also demonstrates the generalization and effectiveness of our method. **We provide more visual comparison in Appendix Sec. I**.

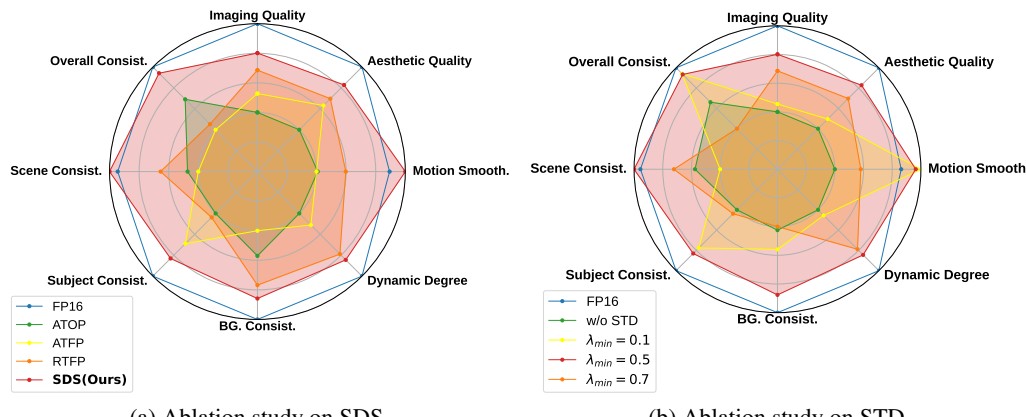

(a) Ablation study on SDS.  (b) Ablation study on STD.

Figure 6: Ablation study of proposed methods on W4A4 CogVideoX-2B.

Table 3: Ablation study on calibration data size.

| Method | Data Size | Calbration Time (Hour) | Imaging Quality | Aesthetic Quality | Overall Consistency |
|---|---|---|---|---|---|
| FP | - | - | 58.61 | 55.25 | 25.91 |
| S$^2$Q-VDiT | 20 | 1.64 | 53.56 | 53.07 | 24.69 |
| S$^2$Q-VDiT | **40** | 2.88 | 55.49 | **53.74** | 25.19 |
| S$^2$Q-VDiT | 80 | 5.56 | **55.52** | 53.64 | **25.21** |

## 4.4 Ablation Study

In Fig. 6, we present ablation studies on Hessian-aware Salient Data Selection (SDS) and Attention-guided Sparse Token Distillation (STD). **To verify the effectiveness of these techniques, we conducted integration experiments with existing PTQ methods in Appendix Sec. E**.

**Ablation on SDS.** We study different calibration data selection methods with our proposed SDS and shown the results in Fig. 6a. We compare three different straightforward methods, including All Timesteps from One Prompt (ATOP), All Timesteps from Five Prompts (ATFP), and Random Timesteps from Five Prompts (RTFP). We selected 40 samples for all methods for fair comparison. We also present the visual comparison in Fig. 3. Our proposed SDS outperforms all other methods in terms of both visual and metric effects while other methods can not maintain high generation quality. **We conducted more ablation experiments on the random seeds and decoupled the two saliences used in SDS. We present the experimental results in Appendix Sec. B**. We further conduct an ablation study on calibration data size in CogVideoX-2B under W4A6 setting and present the results in Tab. 3. It can be seen that the calibration time increases almost linearly with the increase of data size. The performance of 40 data is significantly better than that of 20 data, but the performance improvement of 80 data is minor. Therefore, in the trade-off of performance and calibration time, we choose to use 40 data as the unified experimental settings.

**Ablation on STD.** In Fig. 6b, we compare our proposed STD with no sparse distillation (w/o STD). It can be seen that compared with no STD, all distillation strategies can improve model performance. We also compare different hyperparameters used in Eq (11). We set $\lambda_{max} = 1$ as default and investigate different $\lambda_{min}$ selections which control the relaxation degree on less impactful tokens. It can be seen that all different $\lambda_{min}$ can improve quantization performance which proves the robustness of STD. We select $\lambda_{min} = 0.5$ in the main experiments for the most balanced performance improvement. **We provide more visualization of the sparse patterns in Appendix Sec. H**.

## 4.5 Efficiency Study

We study the deployment efficiency of different-scale video diffusion transformers in Tab. 5. We used the CUDA implementation provided in [62, 47] for deployment and conducted all experiments

on a single NVIDIA A800 GPU. For Inference Memory and Latency, we use a batch size of 1 in Tab 5. Compared with baseline method PTQ4DiT [54], our method brings significant performance improvement with almost no extra inference burden. Compared with FP model, our method can bring $3.94\times$ model memory saving, $1.56\times$ inference memory saving, and $1.28\times$ inference acceleration on CogVideoX-5B. **In Appendix Sec. F, we conducted more experiments on deployment efficiency**.

## 4.6 Calibration Resource Cost

Table 4: Calibration cost on W4A4 CogVideoX-2B.

| Method | GPU Memory (GB) | GPU Time (hour) | Imaging Quality | Aesthetic Quality |
|---|---|---|---|---|
| FP | - | - | 58.61 | 55.25 |
| Q-DiT | 29.85 | 2.69 | 26.26 | 27.66 |
| PTQ4DiT | 33.30 | 2.25 | 20.66 | 28.50 |
| **S²Q-VDiT** | 35.68 | 2.88 | **53.71** | **52.31** |

We reported on the calibration resource consumption of our $S^2$Q-VDiT compared with existing baseline methods Q-DiT [2] and PTQ4DiT [54] in Tab. 4. Compared with existing methods, $S^2$Q-VDiT only increases 2GB memory consumption and 0.2h calibration time, but improves Imaging Quality from 26.26 to 53.71, significantly enhancing the quantization performance. This proves the high efficiency and performance of $S^2$Q-VDiT. **We further reported more detailed calibration resource consumption of each proposed component in Appendix Sec. G**.

Table 5: Efficiency study on different W4A6 models.

| Model | Method | Model Storage (GB) | Inference Memory (GB) | Latency (s) | Imaging Quality | Aesthetic Quality |
|---|---|---|---|---|---|---|
| CogVideoX-5B | FP | 10.375 | 15.801 | 259.2 | 61.80 | 58.88 |
| | PTQ4DiT | 2.633 | 10.139 | 203.1 | 43.54 | 42.70 |
| | **S²Q-VDiT** | **2.633** | 10.145 | 203.2 | **60.75** | **56.90** |
| HunyuanVideo | FP | 23.881 | 29.260 | 191.3 | 62.30 | 62.49 |
| | PTQ4DiT | 6.494 | 13.703 | 175.1 | 48.31 | 50.13 |
| | **S²Q-VDiT** | **6.494** | 13.713 | 175.2 | **58.83** | **59.62** |

## 5 Conclusion

In this paper, we propose $S^2$Q-VDiT, a post-training quantization framework for V-DMs using Salient data and Sparse token distillation. To address the sensitivity to calibration data, we propose Hessian-aware Salient Data Selection to construct high-quality datasets from the perspectives of diffusion and quantization. To address the learning challenge brought by long token sequences, we propose Attention-guided Sparse Token Distillation, which utilizes the natural sparse attention in V-DMs to allocate more loss weights to important tokens. Extensive experiments have shown that $S^2$Q-VDiT outperforms all existing methods on different scales of V-DMs.

## Acknowledgements

This work is partially supported by the National Natural Science Foundation of China under Grant Number 62476264 and 62406312, the Postdoctoral Fellowship Program and China Postdoctoral Science Foundation under Grant Number BX20240385 (China National Postdoctoral Program for Innovative Talents), the Beijing Natural Science Foundation under Grant Number 4244098, the Science Foundation of the Chinese Academy of Sciences, and Swiss National Science Foundation (SNSF) project 200021E_219943 Neuromorphic Attention Models for Event Data (NAMED).

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

# A  Implementation Details

In the main experiment, we use 10 random prompts for generating the candidate calibration samples. We finally selected 40 samples for post-training quantization for all methods. For our method, we use a channel-wise scale used in [55, 62, 54] and a rotation matrix used in [47] for linear quantization. We further use a learnable threshold for clipping the weight and activation min-max value as prior work [30, 18, 47]. We also use GPTQ weight quantizer [13] for our experiment, following prior work [2]. We conduct all the experiments on a single NVIDIA A800 GPU.

For optimization, we train the diag-balancing scale, rotation-based matrix, and learnable clipping threshold following the layer-wise post-training quantization framework as prior works [30, 54]. We use 30 samples and train 15 epochs for each layer. We use AdamW optimizer and cosine learning rate scheduler. For the diag-balancing scale and rotation-based matrix, we use a learning rate of 5e-3. For the learnable clipping threshold, we use a learning rate of 5e-2.

For deployment, we absorb all weight quantization parameters as prior works [54, 55, 62], which brings no extra burden. For activation quantization, we apply online dynamic quantization following [62, 1].

# B  More Ablation on Hessian-aware Salient Data Selection

Table 6: Performance of both 4-bit weight and activation quantization on CogVideoX-2B under three random seeds.

| Method | Imaging Quality | Aesthetic Quality | Motion Smooth. | Dynamic Degree | BG Consist. | Subject Consist. | Scene Consist. | Overall Consist. |
|---|---|---|---|---|---|---|---|---|
| - | 58.69 | 55.25 | 97.95 | 50.00 | 96.40 | 94.30 | 33.79 | 25.91 |
| ATOS | 51.65±(1.76) | 49.79±(0.59) | 98.09±(0.16) | 29.17±(3.40) | 95.82±(0.35) | 93.24±(0.19) | 29.94±(1.35) | 24.31±(0.37) |
| ATDS | 50.63±(0.81) | 50.13±(0.25) | 98.05±(0.11) | 29.63±(2.62) | 95.94±(0.16) | 93.16±(0.41) | 30.98±(2.14) | 24.11±(0.27) |
| DTDS | 50.66±(1.04) | 50.33±(0.19) | 98.03±(0.14) | 31.48±(4.58) | 96.01±(0.16) | 93.07±(0.18) | 30.47±(1.77) | 24.75±(0.25) |
| DS | 52.73±(0.98) | 50.62±(0.81) | 98.15±(0.19) | 31.75±(2.73) | 96.06±(0.18) | 93.29±(0.15) | 31.38±(0.98) | 24.78±(0.22) |
| QS | 52.34±(0.85) | 51.17±(0.23) | 98.11±(0.12) | 32.01±(2.97) | 96.10±(0.17) | 93.57±(0.19) | 31.86±(0.90) | 24.79±(0.23) |
| **SDS(Ours)** | **52.95±(0.69)** | **51.58±(0.11)** | **98.16±(0.09)** | **32.87±(2.36)** | **96.13±(0.15)** | **93.89±(0.17)** | **32.75±(0.77)** | **24.84±(0.26)** |

In this section, we investigate the random seed influence on the quantization performance of different calibration datasets mentioned in Sec. 3.2 and Sec. 4.4. We compare our proposed Hessian-aware Salient Data Selection (SDS) with All Timesteps from One Prompt (ATOP), All Timesteps from Five Prompts (ATFP), and Random Timesteps from Five Prompts (RTFP) using three different random seeds. We further decoupled SDS into Diffusion Salience (DS) in Eq. (3) and Quantization Salience (QS) in Eq. (6) and reported the performance. We present the average results and variance in Tab. 6.

Other straightforward sampling methods have lower average performance and larger variances, proving the influence of random seeds in these random sampling methods. Using our proposed diffusion salience (DS) or quantization salience (QS) can all improve the performance and reduce the impact of random seeds. Only using DS and QS can improve Scene Consistency to over 31 with variances less than 1, while other random sampling methods cannot achieve. By jointly considering two saliences, Hessian-aware Salient Data Selection (SDS) can achieve the best quantization performance with minimal impact from randomness. SDS achieved an average Imaging Quality of 52.95 with only 0.69 variance, while the random sampling only achieved the best average of 51.65 with 1.67 variance.

# C  Detailed Description of Selected Evaluation Metrics

## C.1  VBench Benchmark

For VBench [19] benchmark, we follow the previous work ViDiT-Q [62], which selects 8 dimensions from three key aspects in video-generation task.

**Frame-wise Quality:** In this aspect, we assess the quality of each individual frame without taking temporal quality into concern.

- **Imaging Quality** assesses distortion (e.g., over-exposure, noise) presented in the generated frames using the MUSIQ [23] image quality predictor trained on the SPAQ [7] dataset.

- **Aesthetic Quality** evaluates the artistic and beauty value perceived by humans towards each video frame using the LAION aesthetic predictor [27].

**Temporal Quality:** In this aspect, we assess the cross-frame temporal consistency and dynamics.

- **Dynamic Degree** evaluates the degree of dynamics (i.e., whether it contains large motions) generated by each model.

- **Motion Smoothness** evaluates whether the motion in the generated video is smooth, and follows the physical law of the real world.

- **Subject Consistency** assesses whether the subject's appearance remains consistent throughout the whole video.

- **Background Consistency** evaluate the temporal consistency of the background scenes by calculating CLIP [41] feature similarity across frames.

**Semantics:** In this aspect, we evaluate the video's adherence to the text prompt given by the user. consistency.

- **Scene** evaluates whether the synthesized video is consistent with the intended scene described by the text prompt.

- **Overall Consistency** further use overall video-text consistency computed by ViCLIP [50] on general text prompts as an aiding metric to reflect both semantics and style consistency.

We use three different prompt sets provided by the official github repository of VBench [19] to generate videos. We generate one video for each prompt for evaluation same as ViDiT-Q [62].

- **overall consistency.txt:** includes 93 prompts, used to evaluate overall consistency, aesthetic quality, and imaging quality.

- **subject consistency.txt:** includes 72 prompts, used to evaluate subject consistency, dynamic degree, and motion smoothness.

- **scene.txt:** includes 86 prompts, used to evaluate scene and background consistency.

## C.2 EvalCrafter Benchmark

For EvalCrafter [34] benchmark, consistent with prior work ViDiT-Q [62], we select 5 low-level metrics to evaluate the generation performance.

**CLIPSIM and CLIP-Temp:** CLIPSIM computes the image-text CLIP similarity for all frames in the generated videos, and we report the averaged results. This quantifies the similarity between input text prompts and generated videos. CLIP-Temp computes the CLIP similarity of each two consecutive frames of the generated videos and then gets the averages for each two frames. This quantifies the semantic consistency of generated videos. We use the CLIP-VIT-B/32 [50] model to compute CLIPSIM and CLIP-Temp. We use the implementation from EvalCrafter [34] to compute these two metrics.

**DOVER's VQA:** VQA-Technical measures common distortions like noise, blur, and over-exposure. VQA-Aesthetic reflects aesthetic aspects such as the layout, the richness and harmony of colors, the photo-realism, naturalness, and artistic quality of the frames. We use the Dover [53] method to compute these two metrics.

**FLOW Score:** Flow score was proposed in [34] to measure the general motion information of the video. We use RAFT [48] to extract the dense flows of the video in every two frames, and we calculate the average flow on these frames to obtain the average flow score of each generated video.

We use the prompt sets provided by the official github repository of ViDiT-Q [62] to generate 10 videos for evaluation. We also attached the prompt sets in the supplementary material.

## D  Experiments on more metrics

Following prior work [62], we evaluate different methods on EvalCrafter [34] benchmark for multi-aspects metrics evaluation. We select CLIPSIM, CLIP-Temp, DOVER [53] video quality assessment (VQA) metrics to evaluate the generation quality, and Flow-score to evaluate the temporal consistency. We conduct experiments on CogVideoX-2B, CogVideoX-5B, and HunyuanVideo-13B under W4A6 quantization setting. We present the evaluation results in Tab. 7.

Table 7: Performance of 4-bit weight and 6-bit activation quantization on text-to-video generation under EvalCrafter benchmark. Higher (↑) metrics represent better performance.

| Model | Method | CLIPSIM | CLIP-Temp | VQA-Aesthetic | VQA-Technical | FLOW Score. |
|---|---|---|---|---|---|---|
| CogVideoX-2B | FP | 0.1844 | 0.9978 | 76.64 | 85.02 | 3.452 |
| | Q-DiT | 0.1787 | 0.9978 | 63.15 | 67.37 | 2.331 |
| | PTQ4DiT | 0.1772 | **0.9985** | 58.76 | 52.60 | 1.837 |
| | SmoothQuant | 0.1762 | 0.9981 | 55.18 | 53.87 | 1.378 |
| | Quarot | 0.1808 | 0.9975 | 51.83 | 56.79 | 2.867 |
| | ViDiT-Q | 0.1812 | 0.9976 | 53.09 | 59.84 | 3.040 |
| | **S²Q-VDiT** | **0.1838** | 0.9979 | **70.50** | **73.31** | **3.122** |
| CogVideoX-5B | FP | 0.1814 | 0.9982 | 78.87 | 73.17 | 4.536 |
| | Q-DiT | **0.1835** | 0.9976 | 47.96 | 46.72 | 2.967 |
| | PTQ4DiT | 0.1789 | 0.9984 | 22.93 | 44.07 | 2.230 |
| | SmoothQuant | 0.1742 | 0.9976 | 3.05 | 14.13 | 1.026 |
| | Quarot | 0.1805 | 0.9983 | 33.10 | 43.67 | 3.040 |
| | ViDiT-Q | 0.1795 | 0.9980 | 42.01 | 48.59 | 1.850 |
| | **S²Q-VDiT** | 0.1819 | **0.9987** | **73.45** | **74.41** | **3.688** |
| HunyuanVideo | FP | 0.1910 | 0.9985 | 80.66 | 63.51 | 1.674 |
| | Q-DiT | 0.1871 | **0.9987** | 56.45 | 43.17 | 1.482 |
| | PTQ4DiT | 0.1786 | 0.9973 | 42.17 | 33.69 | 1.089 |
| | SmoothQuant | 0.1782 | 0.9978 | 7.24 | 0.42 | 0.111 |
| | Quarot | 0.1873 | 0.9977 | 66.49 | 52.81 | 0.899 |
| | ViDiT-Q | 0.1895 | 0.9978 | 66.23 | 51.35 | 0.897 |
| | **S²Q-VDiT** | **0.1902** | 0.9985 | **77.80** | **66.38** | **1.562** |

It can be seen that under the EvalCrafter [34] benchmark, our S²Q-VDiT still achieved almost lossless performance and showed significant performance improvement compared to all comparison methods. Especially in terms of VQA-Technical metrics, our S²Q-VDiT even outperforms the full precision model on CogVideoX-5B and HunyuanVideo, while other methods show notable performance degradation. For CogVideoX-5B, S²Q-VDiT achieves 74.41 in VQA-Technical which outperforms the full precision model of 73.17, while current methods achieve the best of 48.59.

## E  Integration with Existing PTQ Methods

The techniques that we proposed Hessian-aware Salient Data Selection (SDS) and Attention-guided Sparse Token Distillation (STD) can also be applied to existing block-wise optimization-based post-training quantization methods. To verify the generality of these two techniques, we combined them with the existing baseline method PTQ4DiT [54] and reported the performance improvement of these techniques on W4A6 CogVideoX-2B under VBench [19] benchmark in Tab. 8. By using the calibration constructed by SDS, we further improved the performance of PTQ4DiT and increased Aesthetic Quality by 1.4. This demonstrates the improvement of SDS-constructed datasets under different optimization frameworks. From optimization perspective, we further improved the Aesthetic Quality to 47.27 by using sparse distillation STD. This also demonstrates the effectiveness and generalization of our attention-based optimization method.

Table 8: Performance of 4-bit weight and 6-bit activation quantization on CogVideoX-2B under VBench evaluation benchmark suite

| Method | Imaging Quality | Aesthetic Quality | Motion Smooth. | Dynamic Degree | BG Consist. | Subject Consist. | Scene Consist. | Overall Consist. |
|---|---|---|---|---|---|---|---|---|
| FP | 58.69 | 55.25 | 97.95 | 50.00 | 96.40 | 94.30 | 33.79 | 25.91 |
| PTQ4DiT | 42.91 | 45.49 | 98.48 | 5.56 | 95.65 | 92.85 | 17.88 | 21.15 |
| +SDS | 43.06 | 46.89 | 98.64 | 11.11 | 95.79 | 93.33 | 18.10 | 22.27 |
| +STD | 43.08 | 47.27 | 98.78 | 9.72 | 95.97 | 93.68 | 19.04 | 22.09 |

# F  More Experiments on Deployment Efficiency

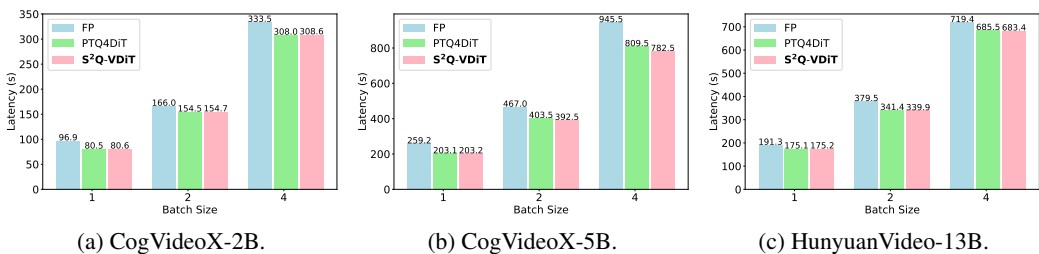

(a) CogVideoX-2B.  (b) CogVideoX-5B.  (c) HunyuanVideo-13B.

Figure 7: Deployment latency comparison under different batch size.

We further expanded the experiments provided in Sec. 4.5. We compared the deployment efficiency of different models under different batch sizes in Fig. 7. Our S²Q-VDiT can bring consistent inference acceleration to different models under different batch sizes. Under the 50-step inference setting of CogVideoX-5B with a batch size of 4, our S²Q-VDiT can reduce the inference latency from 945.4s to 782.5s, achieving a significant acceleration of 163 seconds and outperforming the baseline method PTQ4DiT [54].

Table 9: Calibration cost about each component.

| | Hessian Approximation | | | Attention Computation | |
|---|---|---|---|---|---|
| Method | Construct Time (mins) | Imaging Quality | Method | Calibration Time (hours) | Imaging Quality |
| CogVideoX-2B | | | | | |
| FP | - | 58.69 | FP | - | 58.69 |
| w/o Hessian | 7.708 | 53.16 | w/o Attention | 2.82 | 52.16 |
| w Hessian | 7.717 | **55.49** | w Attention | 2.84 | **55.49** |
| CogVideoX-5B | | | | | |
| FP | - | 61.80 | FP | - | 61.80 |
| w/o Hessian | 20.719 | 58.91 | w/o Attention | 3.97 | 58.23 |
| w Hessian | 20.734 | **60.75** | w Attention | 4.00 | **60.75** |
| HunyuanVideo-13B | | | | | |
| FP | - | 62.30 | FP | - | 62.30 |
| w/o Hessian | 19.505 | 57.25 | w/o Attention | 5.70 | 56.94 |
| w Hessian | 19.508 | **58.83** | w Attention | 5.73 | **58.83** |

# G  More Detailed Calibration Resource Cost

We reported the time increase caused by using the Hessian approximation when constructing the calibration dataset and the attention scores calculation across different scale video generation models in Tab. 9.

It can be seen that the computational burden of using Hessian approximation is minor, but it can bring significant performance improvement. We use the Levenberg-Marquardt approximation [13] to calculate the Hessian approximation, which requires only one step matrix multiplication to obtain the approximate result, and is very efficient.

Also, during the calibration process, we only need to use the Full-Precision model to conduct a single forward calculation of attention scores for all data in advance. When optimizing the quantization model, we can directly get the pre-computed attention scores by the data index, which brings minimal burden.

# H  More Visualization about Sparse Attention Pattern

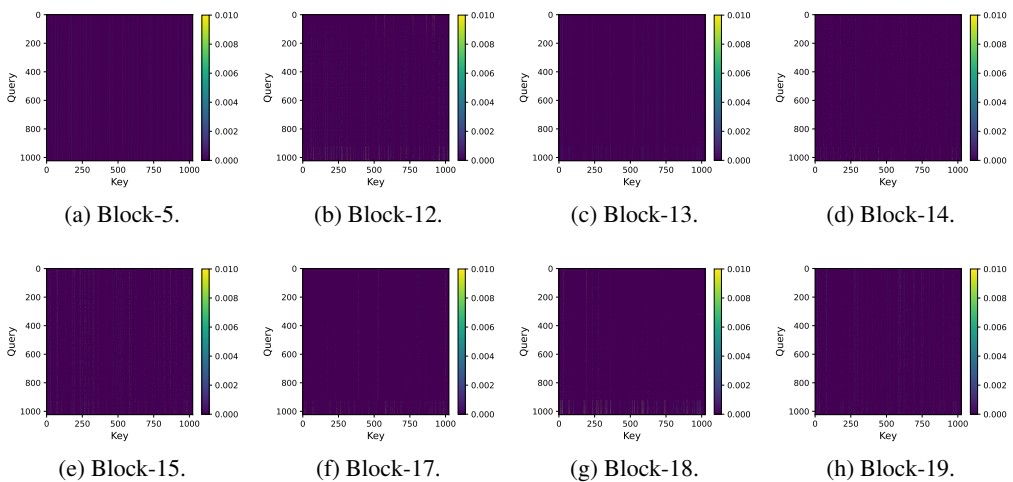

|          |            |            |            |
|----------|------------|------------|------------|
| (a) Block-5. | (b) Block-12. | (c) Block-13. | (d) Block-14. |
| (e) Block-15. | (f) Block-17. | (g) Block-18. | (h) Block-19. |

Figure 8: Visualization of attention heatmaps in CogVideoX-2B.

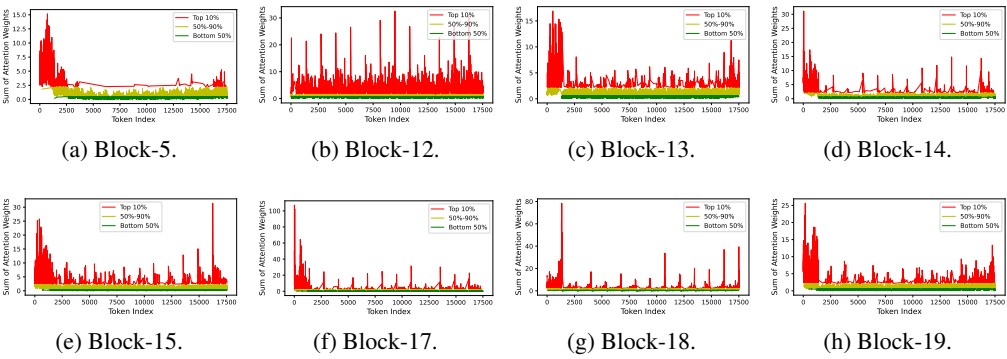

|          |            |            |            |
|----------|------------|------------|------------|
| (a) Block-5. | (b) Block-12. | (c) Block-13. | (d) Block-14. |
| (e) Block-15. | (f) Block-17. | (g) Block-18. | (h) Block-19. |

Figure 9: Visualization of token-wise attention distribution in CogVideoX-2B.

We demonstrate the sparse attention patterns existing in V-DMs that we mentioned in Sec 3.3. We present more visualization results of different blocks of CogVideoX-2B in Fig. 8 and Fig. 9. There is a considerable degree of sparse attention patterns in the most layers of the model, and almost all 90% tokens have significantly lower attention weights than the top 10% tokens. This indicates that

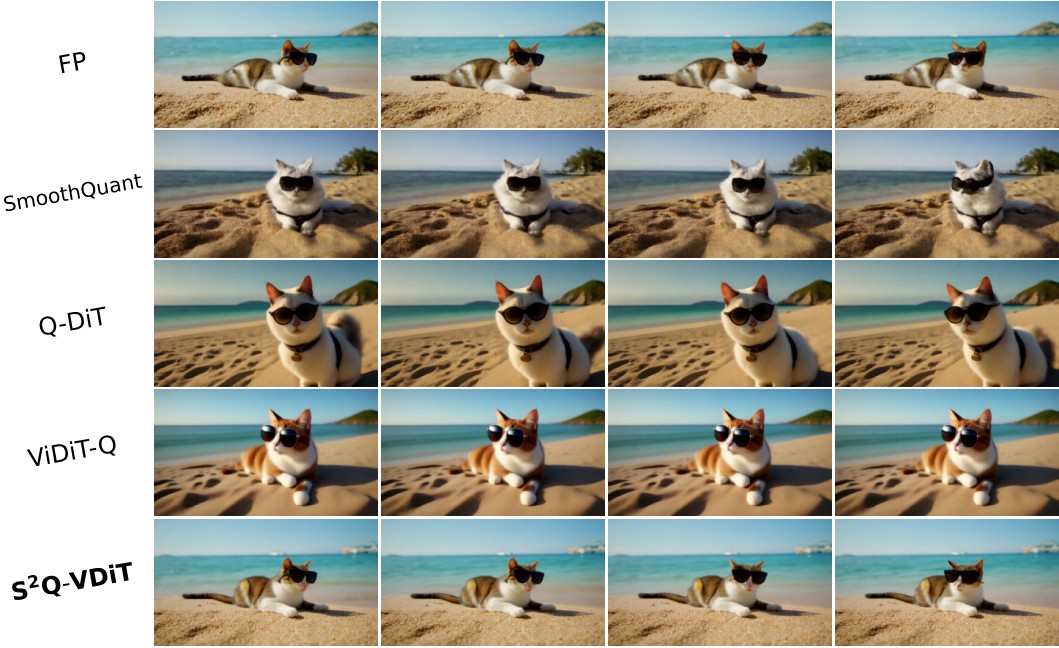

Figure 10: HunyuanVideo-13B results. Prompt: A cat wearing sunglasses on a beach.

sparse attention is commonly present in V-DMs, and almost every layer only has a small portion of tokens that play an important role in the final output. This proves the universality of our observations in Sec. 3.3 and the effectiveness of our Attention-guided Sparse Token Distillation.

# I    More Visualization Results

We present more visual comparison results on HunyuanVideo-13B [24], CogVideoX-5B, and CogVideoX-2B [58] under W4A6 quantization in the following figures. Compared with current methods SmoothQuant [55], Q-DiT [2], ViDiT-Q [62], our S$^2$Q-VDiT made notable visual improvement on different scale video diffusion models. This proves that our S$^2$Q-VDiT not only surpasses existing methods in terms of evaluation metrics but also shows significant improvement in visual effects, demonstrating the effectiveness of our S$^2$Q-VDiT.

# J    Limitations

Although our S$^2$Q-VDiT outperforms existing methods, it cannot achieve completely lossless performance under the most difficult fully 4-bit quantization. We hope to optimize the quantization performance under low bit settings in the future.

# K    Broader Impacts

Our quantized model may be used by people to generate false content, and we will require users to apply our model in legitimate and reasonable scenarios and label it as AI-generated.

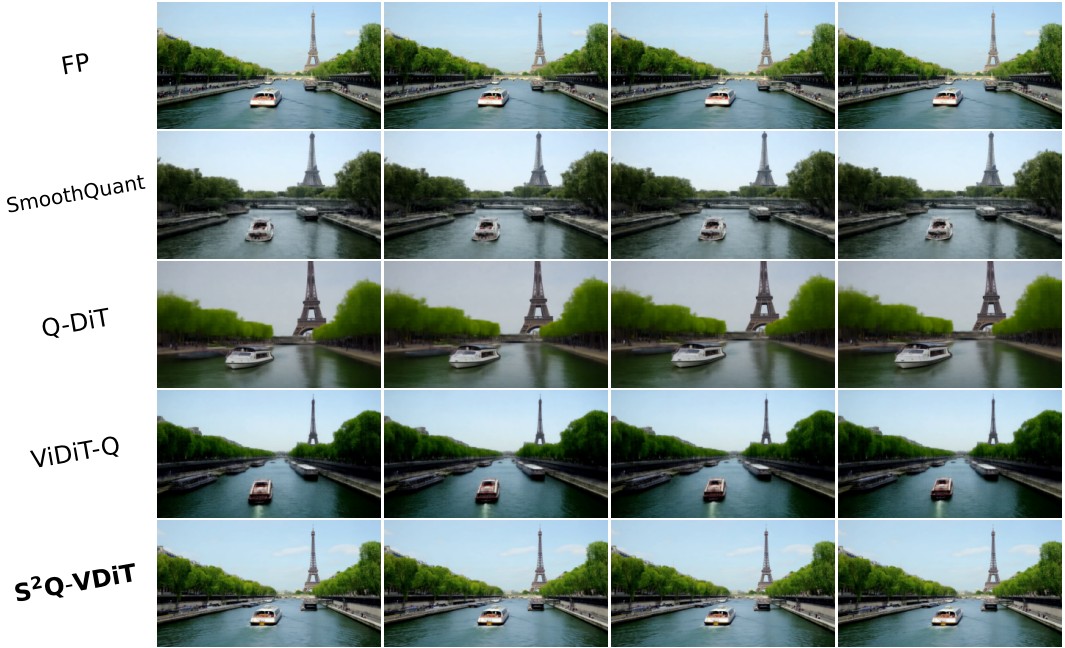

Figure 11: HunyuanVideo-13B results. Prompt: A boat sailing leisurely along the Seine River with the Eiffel Tower in background.

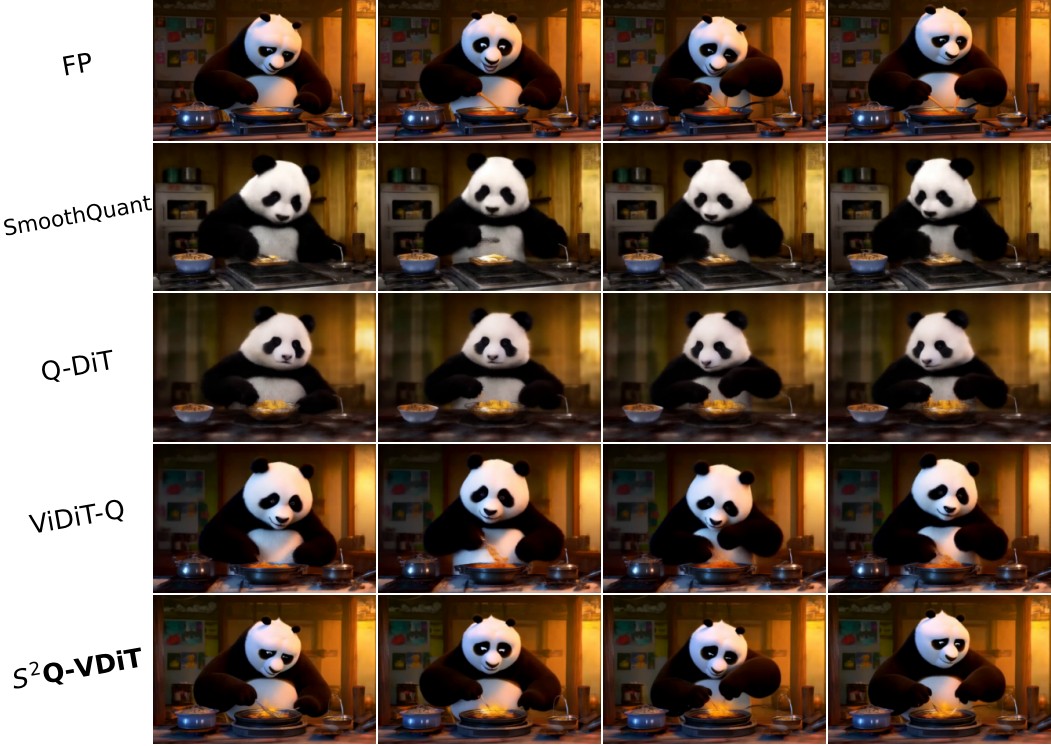

Figure 12: HunyuanVideo-13B results. Prompt: A panda cooking in the kitchen.

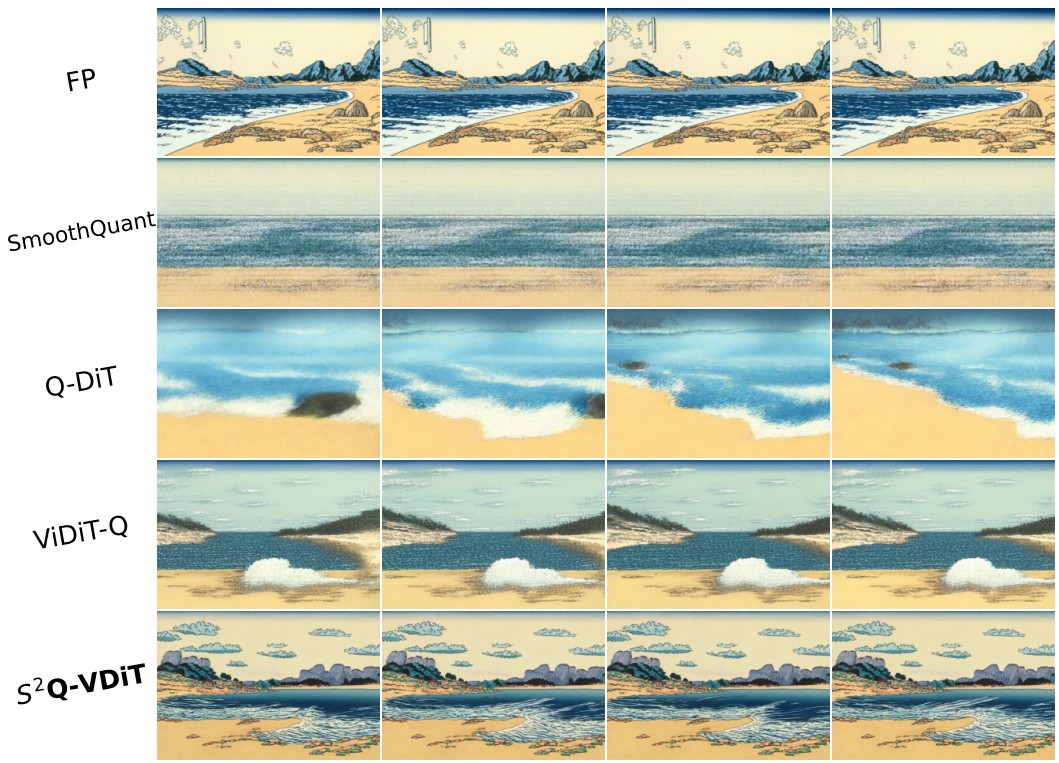

Figure 13: CogVideoX-5B results. Prompt: A beautiful coastal beach in spring, waves lapping on sand by Hokusai, in the style of Ukiyo.

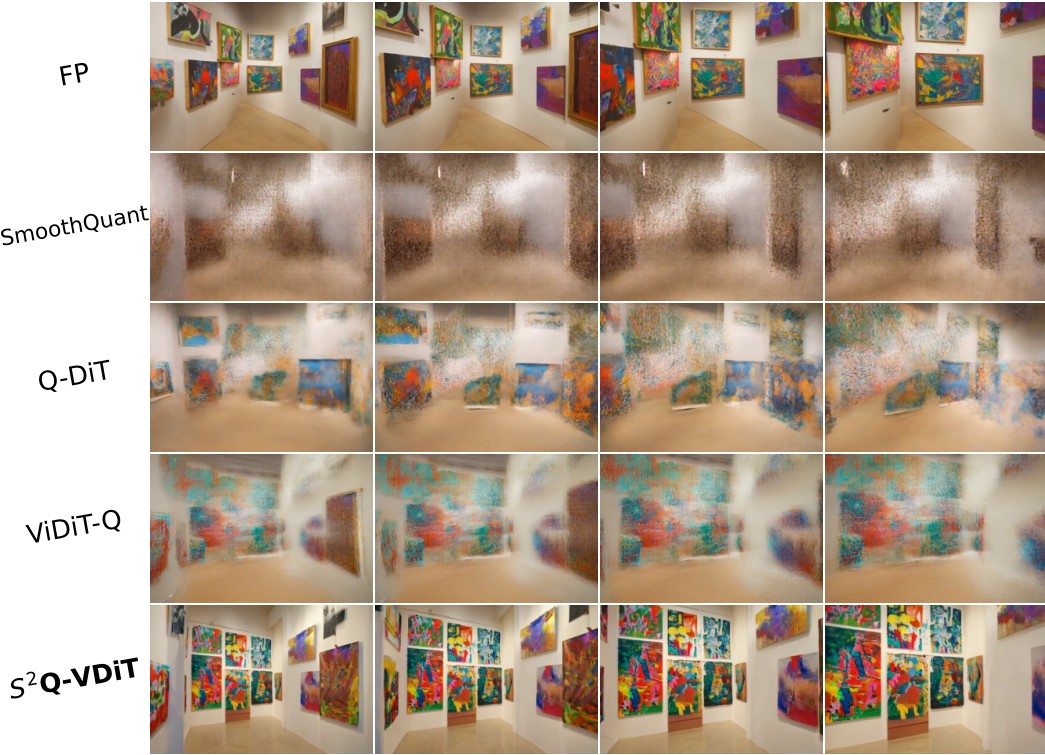

Figure 14: CogVideoX-5B results. Prompt: A modern art museum, with colorful paintings.

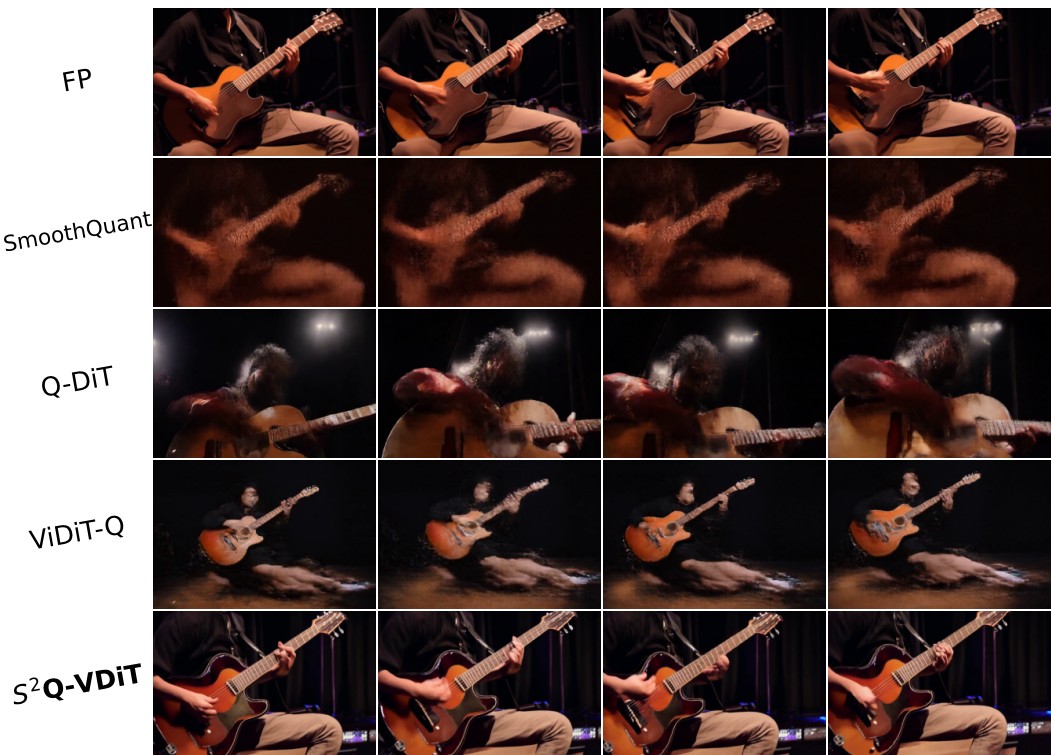

Figure 15: CogVideoX-5B results. Prompt: Yoda playing guitar on the stage.

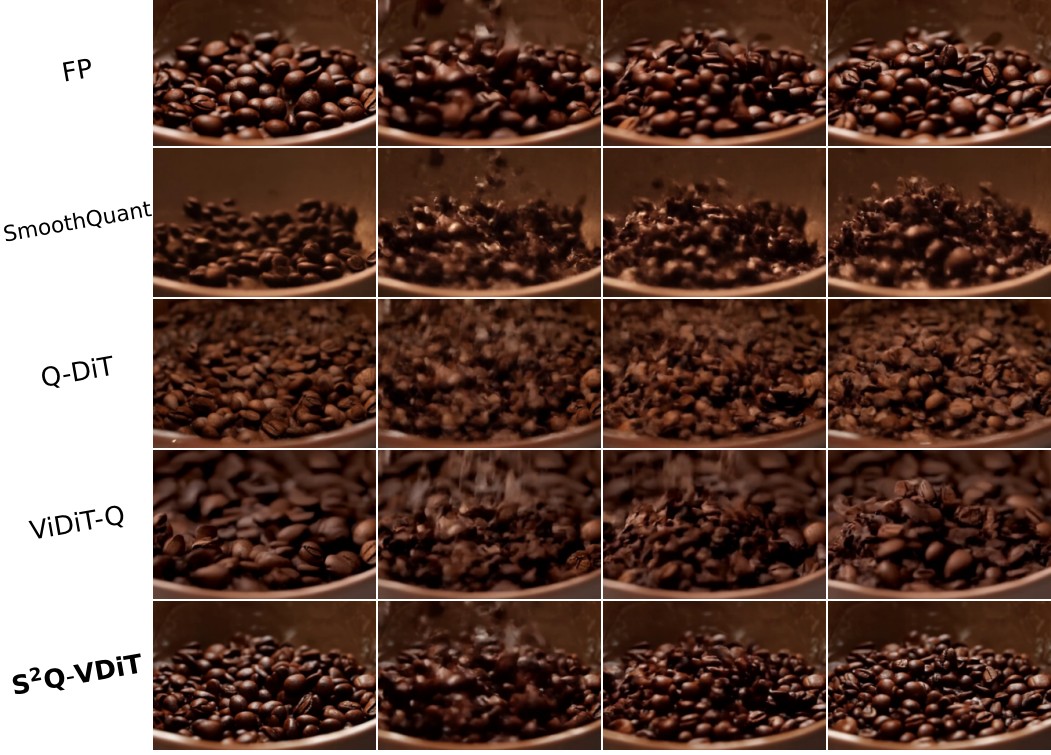

Figure 16: CogVideoX-2B results. Prompt: Macro slo-mo. Slow motion cropped closeup of roasted coffee beans falling into an empty bowl.

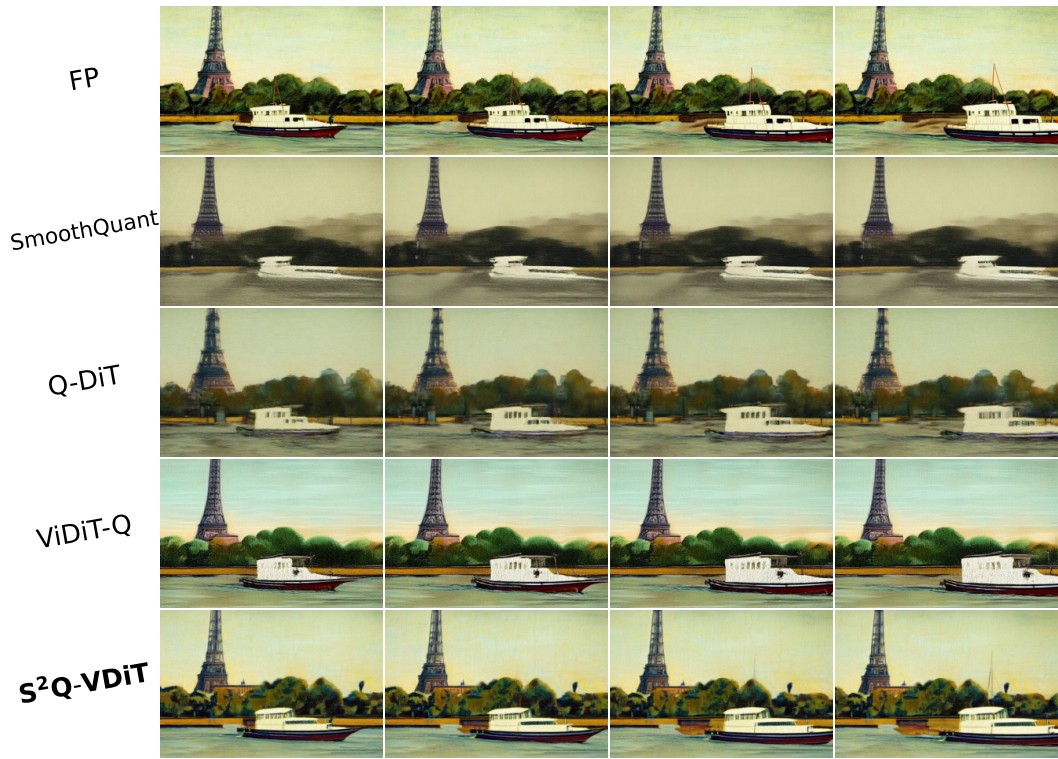

Figure 17: CogVideoX-2B results. Prompt: A boat sailing leisurely along the Seine River with the Eiffel Tower in background by Vincent van Gogh.

