# OpenReview forum: "$\text{S}^2$Q-VDiT: Accurate Quantized Video Diffusion Transformer with Salient Data and Sparse Token Distillation"
_NeurIPS.cc/2025/Conference — NeurIPS 2025 poster_

### Official Review · Reviewer_gBkv · 2025-06-29

**Clarity:** 3
**Significance:** 3
**Originality:** 3
**Rating:** 5
**Confidence:** 4

**Summary:**

This paper proposes S$^2$Q-VDiT, a post-training quantization framework for video diffusion models, which leverages salient data selection and sparse token distillation to improve quantization performance. Specifically, the authors introduce a Hessian-aware salient data selection method to construct a high-quality calibration set, and propose attention-guided sparse token distillation to retain tokens that are more influential to the model’s output. Comprehensive experiments on the VBench benchmark demonstrate the effectiveness of the proposed techniques.

**Questions:**

1. In the W4A6 setting, why are activations quantized to 6 bits while weights use only 4 bits? Some rationale or empirical justification would help clarify this design choice.

2. In the attention modules, do the authors distinguish between cross-attention and self-attention mechanisms?

**Ethical Concerns:**

["NO or VERY MINOR ethics concerns only"]

**Final Justification:**

I am satisfied with the authors’ response and have adjusted my score accordingly.

**Limitations:**

While the proposed methods show strong empirical performance, they may introduce notable computational overhead. In particular, computing the Hessian matrix for salient data selection and attention scores for sparse token distillation could be costly, especially in large-scale or resource-constrained settings.

**Quality:**

3

**Strengths And Weaknesses:**

Strengths:

1. The paper is generally well written and easy to follow.

2. Both the Hessian-aware salient data selection and attention-guided sparse token distillation components are well motivated. The authors provide insightful observations and analyses that support the design of their proposed methods.

3. The experimental evaluation on VBench is thorough and shows promising performance improvements over existing baselines.

Weaknesses:
1. In Figure 5, it would be helpful to include frame indices for better interpretation of temporal consistency.

2. The abbreviation “FP” likely refers to “full precision,” but this should be explicitly defined in the text for clarity.

3. The mathematical derivation from Equation (4) to Equation (6) is somewhat condensed and would benefit from additional explanation or intermediate steps to improve readability and reproducibility.

4. The computation of the Hessian matrix for dataset samples and attention scores for each token may introduce significant computational overhead, especially for large-scale video diffusion models. A discussion of the method’s scalability and computational cost would be valuable.

---

> ### Author Rebuttal · Authors · 2025-07-29
>
> We thank you for your positive feedback and comments. We respond to the concerns below:
>
> >**Q1:** Writing improvement about Fig.5, "FP", and Eq.(4) to Eq.(6).
>
> **A1:** Thank you for your valuable suggestions. We will modify our statements according to your requirements in the revised version to make them clearer and more readable.
>
> >**Q2:** Computational overhead about the Hessian approximation.
>
> **A2:** We report the computational burden of Hessian approximation across different scale video generation models. **In fact, the Hessian approximation is very efficient, and will only bring minor computational burden ($\le$ 0.1%).**
>
> We reported the time increase caused by using the Hessian approximation when constructing the calibration dataset in the following table:
>
> |Method|Construct Time (mins)$\downarrow$|Imaging Quality$\uparrow$|
> |-|-|-|
> CogVideoX-2B
> |FP|-|58.69|
> |Ours (w/o Hessian)|7.708|53.16|
> |Ours (w Hessian)|7.717 (**+0.1％**)|**55.49**|
> CogVideoX-5B
> |FP|-|61.80|
> |Ours (w/o Hessian)|20.719|58.91|
> |Ours (w Hessian)|20.734 (**+0.07％**)|**60.75**|
> HunyuanVideo-13B
> |FP|-|62.30|
> |Ours (w/o Hessian)|19.505|57.25|
> |Ours (w Hessian)|19.508 (**+0.01％**)|**58.83**|
>
> It can be seen that the computational burden of using Hessian approximation is minor, but it can bring significant performance improvement. **We use the Levenberg-Marquardt approximation [1][2] to calculate the Hessian approximation, which requires only one step matrix multiplication to obtain the approximate result, and is very efficient.**
>
> Reference:
>
> [1]. Frantar, et al. Optimal brain compression: A framework for accurate post-training quantization and pruning. NeurIPS 2022.
>
> [2]. Frantar, et al. Gptq: Accurate post-training quantization for generative pre-trained transformers. ICLR 2023.
>
>
> >Q3: Computational overhead about the attention scores calculation.
>
> **A3:** We report the computational burden of the attention scores calculation across different scale video generation models. **The attention scores calculation is very efficient, and will only bring minor computational burden ($\le$ 0.7%).**
>
> We reported the time increase caused by calculating the attention scores during calibration process in the following table:
>
> |Method|Calibration Time (h)$\downarrow$|Imaging Quality$\uparrow$|
> |-|-|-|
> CogVideoX-2B
> |FP|-|58.69|
> |Ours (w/o Attention)|2.82|52.16|
> |Ours (w Attention)|2.84 (**+0.7％**)|**55.49**|
> CogVideoX-5B
> |FP|-|61.80|
> |Ours (w/o Attention)|3.97|58.23|
> |Ours (w Attention)|4.00 (**+0.7％**)|**60.75**|
> HunyuanVideo-13B
> |FP|-|62.30|
> |Ours (w/o Attention)|5.70|56.94|
> |Ours (w Attention)|5.73 (**+0.5％**)|**58.83**|
>
> During the calibration process, **we only need to use the Full-Precision model to conduct a single forward calculation of attention scores for all data in advance.** When optimizing the quantization model, we can directly get the pre-computed attention scores by the data index, which brings minimal burden.
>
> >**Q4:** In the W4A6 setting, why are activations quantized to 6 bits while weights use only 4 bits? Some rationale or empirical justification would help clarify this design choice.
>
> **A4:** Since our main baseline method ViDiT-Q [1] has explored the high bit quantization of the video generation model (e.g., W8A8 and W6A6) and almost achieved the lossless performance, **continuing to experiment under high bit cannot fully reflect the differentiation.** Also, lower bits can bring greater model compression effect and fewer bit operations, which is more significant for the model compression and acceleration. Therefore, **we target the 4-bit weighted quantization that can bring higher compression ratio but is still heavily underexplored.**
>
> We first explored W4A4, a full 4-bit quantization setting. This is a very radical setting for Post-training Quantization, but our method still made great improvement compared with existing methods (See Tab. 2, Line 182) with only minimal performance drop (~1%).
>
> We then hope to guarantee the lossless model performance under 4-bit weight quantization so that it can be widely used in practice. We found that under W4A6 setting, the existing quantization methods still show significant performance degradation, but our method can guarantee almost lossless performance. Therefore, we further conduct W4A6 quantization experiments in Tab.1 (See Line 164). **We further reduce the quantization bit of the lossless video generation model to W4A6 and set a new benchmark for video model quantization.**
>
> Reference:
>
> [1]. Zhao, et al. Vidit-q: Efficient and accurate quantization of diffusion transformers for image and video generation. ICLR 2025.
>
>
> >**Q5:** Difference between cross-attention and self-attention mechanisms.
>
> **A5:** The experiment of CogvideoX and HunyuanVideo in this paper use a single 3D-attention on the architecture, so there is no need to distinguish attention mechanisms. For models that distinguish attention mechanisms, such as Wan2.1 [1], we only use the results of self-attention. We give the detailed explanation in the following paragraph. And **we will add the explanation of this part in the revised version.**
>
> At present, the mainstream open-source video generation models are mainly divided into two types of architectures: MMDiT (e.g., CogVideo-X [2] and HunyuanVideo [3], which are used in the paper's experiments) and DiT (e.g., Wan2.1 [1]) architectures.
>
> **MMDiT architecture uses only a single 3D-Full Attention, which directly applies global attention to visual and text tokens.** Therefore, we directly use the results of 3D attention to calculate the attention score in the proposed *Attention-guided Sparse Token Distillation* (introduced in Sec. 3.3). This is also the implementation in the paper's experimental part.
>
> The DiT architecture applies two different attention mechanisms, self-attention and cross-attention. However, in the current mainstream architecture (e.g., Wan2.1 [1]), text tokens are only injected as fixed conditions, meaning that the output token of the attention block only contains visual tokens. **The purpose of our proposed *Attention-guided Sparse Token Distillation* is to reweight the output token's contribution in the alignment to the Full-Precision model. Therefore, we only use the attention score of self-attention mechanism to calculate token-wise distribution mentioned in Eq. 11 (Line 178).**
>
> Reference:
>
> [1]. Wan, et al. Wan: Open and advanced large-scale video generative models. Arxiv 2025.
>
> [2]. Yang, et al. Cogvideox: Text-to-video diffusion models with an expert transformer. Arxiv 2024.
>
> [3]. Kong, et al. Hunyuanvideo: A systematic framework for large video generative models. Arxiv 2024.

---

> > ### Comment · Reviewer_gBkv · 2025-08-03
> >
> > Thank you for the authors' detailed and thoughtful response. In my opinion, the authors have addressed most of my concerns and clarified the key points I raised. I would like to follow the ongoing discussion between the authors and other reviewers before making my final decision.

---

> > > ### Author Response · Authors · 2025-08-04
> > >
> > > Dear Reviewer gBkv,
> > >
> > > Thank you once again for your recognition of our work and responses! We are glad to learn that our rebuttal has largely addressed your concerns.
> > >
> > > Your discussion has greatly improved our work, and we will continue to add the discussed content in the upcoming revised version. If you have any additional questions, please feel free to ask, and we will be glad to answer them for you.
> > >
> > > Best regards,
> > >
> > > Authors of Paper 14993

---

### Official Review · Reviewer_ZQ33 · 2025-06-30

**Clarity:** 3
**Significance:** 3
**Originality:** 3
**Rating:** 4
**Confidence:** 3

**Summary:**

This paper introduces S²Q-VDiT, a post-training quantization (PTQ) framework designed specifically for video diffusion transformers (V-DMs). The core contribution addresses two major challenges in quantizing V-DMs: the high variance in quantization performance due to limited calibration data, and the inefficiency of uniform token treatment during optimization in the presence of long token sequences and sparse attention patterns. To tackle these, the authors propose two key components: Hessian-aware Salient Data Selection (SDS) and Attention-guided Sparse Token Distillation (STD).

**Questions:**

please refer to the weaknesses part.

**Ethical Concerns:**

["NO or VERY MINOR ethics concerns only"]

**Final Justification:**

Thank you to the authors for their detailed rebuttal. I have reviewed both their responses and the comments from the other reviewers. The rebuttal satisfactorily addressed most of my concerns. Therefore, I will maintain my original score.

**Limitations:**

yes

**Quality:**

3

**Strengths And Weaknesses:**

Strengths:

1. The paper identifies and addresses crucial challenges specific to post-training quantization of Video Diffusion Models (V-DMs), particularly the high variance from limited calibration data and inefficient token treatment in long sequences.

2. The proposed Hessian-aware Salient Data Selection (SDS) and Attention-guided Sparse Token Distillation (STD) are well-motivated and technically sound. SDS combines diffusion informativeness and quantization sensitivity effectively. STD leverages the empirically observed sparse attention patterns in V-DMs for more efficient optimization.

3. The methods are thoroughly evaluated on large-scale V-DMs ranging from 2B to 13B parameters, demonstrating consistent state-of-the-art performance against various baselines. The evaluation uses the VBench benchmark suite, which covers multiple quality dimensions including Imaging Quality, Aesthetic Quality, Motion Smoothness, Dynamic Degree, and various consistencies.

Weaknesses:

1. While the paper mentions that QAT requires extensive training time and large datasets, a more direct comparison or discussion on scenarios where QAT might still be preferred could strengthen the argument for PTQ.

2. Although the paper highlights a 1.3x inference acceleration , the detailed latency numbers in Table 3 show a more modest improvement, especially for the Hunyuan Video model (191.3s for FP vs. 175.2s for S²c-VDIT). This translates to approximately a 1.09x acceleration for Hunyuan Video and 1.28x for Cog VideoX-5B (259.2s for FP vs. 203.2s for S²-VDIT). Given the significant model compression (3.9x), the observed inference acceleration is not as substantial as might be expected, which could make the quantization less convincing for applications prioritizing runtime speed above all else.

---

> ### Author Rebuttal · Authors · 2025-07-29
>
> Thank you for reviewing our manuscript and providing valuable suggestions. Here are our responses to some of the concerns you raised:
>
> >**Q1:** While the paper mentions that QAT requires extensive training time and large datasets, a more direct comparison or discussion on scenarios where QAT might still be preferred could strengthen the argument for PTQ.
>
> **A1:** QAT conducts full training on the model weight and quantization parameters. which can guarantee relatively good performance under some extreme bits (1-2 bits). However, compared with the PTQ paradigm, QAT usually requires more training resources. For example, **we compare our PTQ method with the existing QAT work for diffusion models [1][2] and report the required training resources:**
>
> |Paradigm|Model Size|Training Time|Training Data|
> |-|-|-|-|
> |QAT|LDM (**~1GB**) [3]|~30h (**10×**)|300k (**7500×**)|
> |PTQ|CogVideoX (**~4GB**)|~3h|40|
>
> **Compared with the QAT paradigm, the PTQ paradigm we use can greatly reduce computing resources and calibrate larger models, which fully reflects its efficiency.** It is still worth mentioning that under extreme bit (1-2 bit) quantization, QAT is still a reliable method that can guarantee the maximum performance of the model. However, in this paper, we mainly explore 4-bit quantization, and only an efficient PTQ process can ensure the nearly lossless model performance.
>
> Reference:
>
> [1]. Zheng, et al. Bidm: Pushing the limit of quantization for diffusion models. NeurIPS 2024.
>
> [2]. Zheng, et al. Binarydm: Accurate weight binarization for efficient diffusion models. ICLR 2025.
>
> [3]. Rombach, et al. High-resolution image synthesis with latent diffusion models. CVPR 2022.
>
> >**Q2:** Although the paper highlights a 1.3x inference acceleration, the detailed latency numbers in Table 3 show a more modest improvement, especially for the Hunyuan Video model (191.3s for FP vs. 175.2s for S²c-VDIT). This translates to approximately a 1.09x acceleration for Hunyuan Video and 1.28x for Cog VideoX-5B (259.2s for FP vs. 203.2s for S²-VDIT). Given the significant model compression (3.9x), the observed inference acceleration is not as substantial as might be expected, which could make the quantization less convincing for applications prioritizing runtime speed above all else
>
> **A2:** Our empirical observation shows that HunyuanVideo does have a relatively low acceleration ratio. However, our quantization method has significant memory compression and inference acceleration across a variety of popular video generation models, and the acceleration effect is more obvious in high-resolution generation.
>
> For the deployed quantization model, compared with CogVideo-X, we do observe that HunyuanVideo has a relatively lower inference acceleration rate. We attribute this to the dual-stream DiT block used in HunyuanVideo. The dual-stream DiT block introduces much more high-precision normalization, scale, and shift operations that cannot fully benefit from quantization. **We also observed that for models that support multi-resolution generation like HunyuanVideo and Wan2.1 [1], higher resolutions that are closer to the business scenarios can bring better acceleration effects.** We present the comprehensive evaluation results here:
>
> |Method|Resolution|Model Storage (GB)$\downarrow$|Inference Memory (GB)$\downarrow$|Latency (s)$\downarrow$|
> |-|-|-|-|-|
> CogVideoX-2B
> |FP|480×720|3.2|6.7|166.5|
> |Ours|480×720|0.8 (**4.0×**)|5.5 (**1.2×**)|97.5 (**1.71×**)|
> CogVideoX-5B
> |FP|480×720|10.4|15.8|259.2|
> |Ours|480×720|2.6 (**3.9×**)|10.1 (**1.6×**)|203.2 (**1.28×**)|
> HunyuanVideo-13B
> |FP|512×768|23.9|29.3|191.3|
> |Ours|512×768|6.5 (**3.7×**)|13.7 (**2.1×**)|175.2 (**1.09×**)|
> |FP|720×1280|23.9|35.8|807.5|
> |Ours|720×1280|6.5 (**3.7×**)|22.7 (**1.6×**)|684.0 (**1.18×**)|
> Wan2.1-14B
> |FP|480×832|26.6|33.9|1202.0|
> |Ours|480×832|7.00 (**3.8×**)|16.1 (**2.1×**)|971.0 (**1.24×**)|
> |FP|720×1280|26.6|42.5|4031.0|
> |Ours|720×1280|7.00 (**3.8×**)|26.0 (**1.6×**)|3160.0 (**1.28×**)|
>
> We can see that latency results are consistent with our observations. Also on the four popular open-source video generation models, quantization can bring continuous inference acceleration. **The benefits of quantization are not only reflected in latency.** The compression of model storage can greatly reduce the model storage occupation. The significant reduction of inference memory can make it more widely used on devices with limited computing resources, which has high research value for the wide application of video generation models.
>
> Reference:
>
> [1]. Wan, et al. Wan: Open and advanced large-scale video generative models. Arxiv 2025.

---

### Official Review · Reviewer_uJYk · 2025-07-01

**Clarity:** 3
**Significance:** 3
**Originality:** 3
**Rating:** 5
**Confidence:** 4

**Summary:**

This paper introduces S2Q-VDiT, a post-training quantization (PTQ) framework designed to address the performance degradation of video diffusion models (V-DMs) under low-bit quantization. Due to the long token sequences and limited calibration budget inherent in V-DMs, existing PTQ approaches often struggle with unstable performance. To tackle this, S2Q-VDiT introduces two main components. The first is Hessian-aware Salient Data Selection, which scores and selects calibration samples based on their contribution to the denoising process and their sensitivity to quantization, using an efficient approximation of the Hessian. The second is Attention-guided Sparse Token Distillation, which leverages the sparsity of attention maps to prioritize important tokens during quantization loss computation, assigning higher weights to those with stronger semantic relevance.
The method is evaluated on large-scale video diffusion models, including CogVideoX and HunyuanVideo, under W4A6 and W4A4 quantization settings. Results show that S2Q-VDiT maintains high generation quality while achieving up to 3.9× model compression and 1.3× inference speedup. Additionally, the proposed techniques can be integrated into other PTQ frameworks and require only a small number of calibration samples, with minimal one-time overhead during deployment.

**Questions:**

Please refer to the weakness part.

**Ethical Concerns:**

["NO or VERY MINOR ethics concerns only"]

**Final Justification:**

Thank you to the authors for responding to my questions and concerns. After reading the author's response and other reviewers' comments, I will raise my score to 5.

**Limitations:**

Yes.

**Paper Formatting Concerns:**

No.

**Quality:**

3

**Strengths And Weaknesses:**

Strength:
* The paper is easy to follow, with a clear structure that helps the reader understand the challenges being addressed and the proposed solutions.
* The authors introduce a Hessian-aware salient data selection method, offering a novel and principled approach for selecting calibration data, which improves the post-training quantization process compared to traditional random sampling.
* The method effectively leverages the sparse attention patterns inherent in video diffusion models (VDMs), emphasizing salient tokens to enhance PTQ performance.
* Extensive experiments are conducted across multiple models and quantization settings to validate the robustness of the proposed approach. The method is compared with various PTQ baselines using multiple evaluation metrics, showing strong performance in generation quality. Furthermore, it outperforms other calibration data selection strategies across all reported metrics.

 Weaknesses:
* It seems that the calculation of the Hessian approximation introduces non-negligible computational overhead, which can become significant as the number of calibration samples or timesteps increases.
* The paper lacks a discussion or ablation study on the effect of calibration data size. It does not investigate how performance varies with different numbers of calibration samples, nor does it explore the trade-off between calibration data size and quantization quality.

---

> ### Author Rebuttal · Authors · 2025-07-29
>
> We are deeply grateful for your support of our work, and we provide detailed responses to your comments as follows:
>
> >**Q1:** It seems that the calculation of the Hessian approximation introduces non-negligible computational overhead, which can become significant as the number of calibration samples or timesteps increases.
>
> **A1:** We reported the time increase caused by using the Hessian approximation when constructing the calibration dataset in different models. **Compared with without using Hessian, using Hessian approximation will only bring less than 0.1% additional time, which is very efficient.**
>
> We provide the detailed comparison in the following table:
> |Method|Construct Time (mins)$\downarrow$|Imaging Quality$\uparrow$|
> |-|-|-|
> CogVideoX-2B (50steps)
> |FP|-|58.69|
> |Ours (w/o Hessian)|7.708|53.16|
> |Ours (w Hessian)|7.717 (**+0.1％**)|**55.49**|
> CogVideoX-5B (50steps)
> |FP|-|61.80|
> |Ours (w/o Hessian)|20.719|58.91|
> |Ours (w Hessian)|20.734 (**+0.07％**)|**60.75**|
> HunyuanVideo-13B (30steps)
> |FP|-|62.30|
> |Ours (w/o Hessian)|19.505|57.25|
> |Ours (w Hessian)|19.508 (**+0.01％**)|**58.83**|
>
> It can be seen that the computational burden of using Hessian approximation is minor, but it can bring significant performance improvement. **We use the Levenberg-Marquardt approximation [1][2] to calculate the Hessian approximation, which requires only one step matrix multiplication (See Eq. 6) to obtain the approximate result, and is very efficient.**
>
> Reference:
>
> [1]. Frantar, et al. Optimal brain compression: A framework for accurate post-training quantization and pruning. NeurIPS 2022.
>
> [2]. Frantar, et al. Gptq: Accurate post-training quantization for generative pre-trained transformers. ICLR 2023.
>
> >**Q2:** The paper lacks a discussion or ablation study on the effect of calibration data size. It does not investigate how performance varies with different numbers of calibration samples, nor does it explore the trade-off between calibration data size and quantization quality.
>
> **A2:** We conduct ablation study on calibration data size in CogVideoX-2B under W4A6 setting. **Increasing data size will bring additional calibration burden, but it does not always bring significant performance improvement. We finally chose a compromise solution of 40 data size.**
>
> We provide the detailed results in the following table:
> |Method|Data Size|Calbration Time (h)|Imaging Quality$\uparrow$|Aesthetic Quality$\uparrow$|Overall Consistency$\uparrow$|
> |-|-|-|-|-|-|
> |FP|-|-|58.69|55.25|25.91|
> |Ours|20|1.64|53.26|53.07|24.69|
> |Ours|40|2.88|55.49|**53.74**|25.19|
> |Ours|80|5.56|**55.52**|53.64|**25.21**|
>
> It can be seen that the calibration time increases almost linearly with the increase of data size. The performance of 40 data is significantly better than that of 20 data, but the performance improvement of 80 data is minor. Therefore, in the trade-off of performance and calibration time, we choose to use 40 data as the unified experimental settings, as we described in Appendix Sec. A (Line 441).

---

> > ### Comment · Reviewer_uJYk · 2025-08-05
> >
> > Thank you to the authors for responding to my questions and concerns. After reading the author's response and other reviewers' comments, I will raise my score to 5.

---

> > > ### Author Response · Authors · 2025-08-05
> > >
> > > Dear Reviewer uJYk,
> > >
> > > Thank you once again for your recognition of our work and responses! We’re excited to hear that our rebuttal has effectively addressed your concerns and that you are willing to raise your recommendation to clear acceptance (5).
> > >
> > > We greatly appreciate your suggestion and will continue to add the discussed content in the revised version. Your insights have been invaluable in improving the quality and clarity of our work.
> > >
> > > Best regards,
> > >
> > > Authors of Paper 14993

---

### Official Review · Reviewer_QBEu · 2025-07-02

**Clarity:** 3
**Significance:** 3
**Originality:** 3
**Rating:** 5
**Confidence:** 5

**Summary:**

This paper proposes a post training quantization method for video diffusion transformer. The paper addresses the issues in PTQ from two perspectives regarding the long sequence characteristics of video DiTs. For limited calibration data caused by long sequences, the paper proposes Hessian-aware Salient Data Selection to select key calibration data. Aiming at the naturally occurring sparse attention phenomenon in V-DiTs,  Attention-guided Sparse Token Distillation is proposed to focus more on key tokens to improve the performance during calibration process. Massive experiments are conducted on multiple popular V-DiTs and achieved SOTA performance.

**Questions:**

1.Providing more low-level metrics for video evaluation like PSNR and SSIM would further strength the superiority of the proposed methods.

2.Providing more acceleration results under different resolution of HunyuanVideo would be better for verifying the acceleration effect in different scenarios.

**Ethical Concerns:**

["NO or VERY MINOR ethics concerns only"]

**Final Justification:**

My concerns are resolved. I remain positive on this work.

**Limitations:**

yes

**Quality:**

3

**Strengths And Weaknesses:**

## Strengths:
1. The paper focuses on the problems caused by long token sequences in video diffusion models during PTQ process, which is a novel and valuable perspective.  And the paper writing is clear and easy to understand, with good definitions and solutions to the problems from both calibration data and optimization process perspective.
2. The method proposed in the paper has a good theoretical basis and a large number of model analyses and references, ensuring the effectiveness of the methods.
3. Compared with existing methods, the paper has significant improvements in metrics and visual effects, proving the effectiveness of the method. And random seeds experiments are provided in the appendix to demonstrate the robustness of the method.
4. The paper conducted extensive quantitative and ablation experiments on multiple large-scale models, demonstrating the generality and effectiveness of the method.

##Weaknesses:
1. The paper provides evaluation results under VBench and Evalcraft benchmark, but providing more low-level metrics such as PSNR and SSIM will more fully demonstrate the effectiveness of the method.

2. Large video diffusion models such as HunyuanVideo support video generation at multiple resolutions. The paper provides acceleration effects under union experiment setting. But providing different resolutions results, especially at higher resolutions, will further illustrates the potential of the method for acceleration.

---

> ### Author Rebuttal · Authors · 2025-07-29
>
> Thank you very much for your high recognition of our work and the valuable suggestions you provided. Our response is as follows:
>
> > **Q1:** Providing more low-level metrics for video evaluation like PSNR and SSIM would further strength the superiority of the proposed methods.
>
> **A1:** We provide more low-level metrics (PSNR, SSIM, and LPIPS) comparison results of CogVideoX-5B and HunyuanVideo under W4A6 setting. **Compared with the existing state-of-the-art methods Quarot [1] and ViDiT-Q [2], our method achieves the best performance in all metrics**.
>
> We provide the results in the following table. The experimental results show that our method not only has better results on the VBench (Tab.1 and Tab.2) and EvalCrafter (Tab. 5) provided in the main paper, but also has advantages on the low-level metrics. It further proves that our method can generally improve the model performance.
>
> |Method|PSNR$\uparrow$|SSIM$\uparrow$|LPIPS$\downarrow$|
> |-|-|-|-|
> |HunyuanVideo||||
> |Quarot|19.13|0.6168|0.3864|
> |ViDiT-Q|19.22|0.6171|0.3837
> |**Ours**|**19.64**|**0.6359**|**0.3068**|
> |CogVideoX-5B|
> |Quarot|16.17|0.4963|0.4780|
> |ViDiT-Q|17.26|0.5389|0.4620|
> |**Ours**|**17.58**|**0.5865**|**0.3322**|
>
> Reference:
>
> [1]. Ashkboos, et al. QuaRot: Outlier-Free 4-Bit Inference in Rotated LLMs. NeurIPS 2024.
>
> [2]. Zhao, et al. Vidit-q: Efficient and accurate quantization of diffusion transformers for image and video generation. ICLR 2025.
>
> > **Q2:** Providing more acceleration results under different resolutions of HunyuanVideo would be better for verifying the acceleration effect in different scenarios.
>
> **A2:** We provide multi-resolution efficiency study on HunyuanVideo under W4A4 setting. Compared with FP model, our method achieves obvious model compression and inference acceleration under various resolutions.
>
> We provide the detailed results in the following table:
>
> |Method|Resolution|Model Storage (GB)$\downarrow$|Inference Memory (GB)$\downarrow$|Latency (s)$\downarrow$|
> |-|-|-|-|-|
> |FP|512×768|23.9|29.3|191.3|
> |**Ours**|512×768|6.5 (**3.7×**)|13.7 (**2.1×**)|175.2 (**1.09×**)|
> |FP|720×1280|23.9|35.8|807.5|
> |**Ours**|720×1280|6.5 (**3.7×**)|22.7 (**1.6×**)|684.0 (**1.18×**)|
>
> **We find that the model acceleration effect is more obvious when the resolution is increased.** We believe that when the resolution is improved and the number of tokens to be processed by the model is increased, the greater computational burden can benefit more from the quantization technique. This fully reflects the great potential of quantization for generating high-resolution video.

---

> > ### Comment · Reviewer_QBEu · 2025-08-05
> >
> > Thanks for the reply. My concerns are well addressed. I remain positive on this work.

---

> > > ### Author Response · Authors · 2025-08-07
> > >
> > > Dear Reviewer QBEu,
> > >
> > > Thank you very much for your thoughtful feedback and for keeping your positive score. We sincerely appreciate your recognition of our work. We will revise the paper to incorporate all of your feedback.
> > >
> > > Best regards,
> > >
> > > Authors of Paper 14993

---

### Note · Authors · 2025-08-12

Dear Area Chair and Reviewers,

# 1. Apperciation

We sincerely thank the reviewers for their **thorough and constructive feedback** throughout the review process. Their insightful comments have been invaluable in helping us improve the quality of our work.

# 2. Response to Reviewer Concerns

We have **addressed all technical concerns** raised by the reviewers. In addition, we revised the original submission to **enhance overall clarity, readability, and the depth of technical and theoretical discussions.** These revisions ensure that the key ideas and contributions are communicated more effectively.

# 3. Contributions and Experimental Analysis

Our paper found the **data sensitivity and sparse attention characteristics of the video generation model**, and designed innovative methods to ensure the high quality of the calibration data and the performance in the optimization process, **significantly improving the quantization performance**. We also conduct extensive experiments that fully **demonstrate the effectiveness of our methods and the actual inference efficiency**. This empirical study provides valuable insights and opens up numerous directions for future research in this important area.

# 4. Closing Remarks

We hope that our efforts and the improvements made are reflected in the evaluation, and we appreciate the opportunity to contribute to this field !!!

Thank you for your consideration.

---

### Decision · Program_Chairs · 2025-09-17

**Decision:**

Accept (poster)

**Comment:**

The paper considers the problem of post-training quantization in video diffusion transformers (V-DMs) where availability of calibration data would be much less than in image-diffusion models. To this end, the paper leverages two ideas, a Hessian-based salient data selection that ranks diffusion time steps based on the generated content, and an attention-guided sparse token distillation leveraging the observation that only a small number of tokens are actually attended in the diffusion steps. Experiments are presented on text-to-video methods using VBench and EvalCrafter, showing clear improvements in image quality and dynamic degree.

The paper received four reviews, all reviewers very positive in their recommendation. There were concerns raised on a few aspects.
1) Missing experiments, e.g., with different resolutions, evaluation metrics, calibration data size, etc. (QBEu, uJYk) -- authors provided the numbers during rebuttal.
2) Computational overhead from the Hessian (uJYk, ZQ33, gBkv) -- clarifications were provided during the rebuttal.

AC had an independent reading of the paper and agrees with the reviewers that the paper studies an important problem in video diffusion and makes a practical contribution towards enabling video diffusion models for low-resource settings. Thus, recommends acceptance. Authors should incorporate the additional results into the revised paper.